# A novel SNF2 ATPase complex in *Trypanosoma bruceí* with a role in H2A.Z-mediated chromatin remodelling

Tim Vellmer[1], Laura Hartleb[1], Albert Fradera Sola[2], Susanne Kramer[1], Elisabeth Meyer-Natus[1], Falk Butter[2]*, Christian J. Janzen[1]*

1 Department of Cell & Developmental Biology, Biocenter, University of Würzburg, Würzburg, Germany,
2 Quantitative Proteomics, Institute of Molecular Biology (IMB), Mainz, Germany

* f.butter@imb-mainz.de (FB); christian.janzen@uni-wuerzburg.de (CJJ)

## Abstract

A cascade of histone acetylation events with subsequent incorporation of a histone H2A variant plays an essential part in transcription regulation in various model organisms. A key player in this cascade is the chromatin remodelling complex SWR1, which replaces the canonical histone H2A with its variant H2A.Z. Transcriptional regulation of polycistronic transcription units in the unicellular parasite *Trypanosoma brucei* has been shown to be highly dependent on acetylation of H2A.Z, which is mediated by the histone-acetyltransferase HAT2. The chromatin remodelling complex which mediates H2A.Z incorporation is not known and an SWR1 orthologue in trypanosomes has not yet been reported. In this study, we identified and characterised an SWR1-like remodeller complex in *T. brucei* that is responsible for Pol II-dependent transcriptional regulation. Bioinformatic analysis of potential SNF2 DEAD/Box helicases, the key component of SWR1 complexes, identified a 1211 amino acids-long protein that exhibits key structural characteristics of the SWR1 subfamily. Systematic protein-protein interaction analysis revealed the existence of a novel complex exhibiting key features of an SWR1-like chromatin remodeller. RNAi-mediated depletion of the ATPase subunit of this complex resulted in a significant reduction of H2A.Z incorporation at transcription start sites and a subsequent decrease of steady-state mRNA levels. Furthermore, depletion of SWR1 and RNA-polymerase II (Pol II) caused massive chromatin condensation. The potential function of several proteins associated with the SWR1-like complex and with HAT2, the key factor of H2A.Z incorporation, is discussed.

## Author summary

*Trypanosoma brucei* is the causative agent of African trypanosomiasis (sleeping sickness) in humans and nagana in cattle. Its unusual genomic organisation featuring large polycistronic units requires a general mechanism of transcription initiation, because individual gene promoters are mostly absent. Despite the fact that the histone variant H2A.Z has previously been identified as a key player of transcription regulation, the complex responsible for correct H2A.Z incorporation at transcription start sites (TSS) remains elusive. In other

**Data Availability Statement:** All relevant data are within the manuscript and its Supporting Information files.

**Funding:** CJJ was funded by a research grant of the Deutsche Forschungsgemeinschaft (DFG, JA 1013/4-1). The funders had no role in study design, data collection and analysis, decision to publish, or preparation of the manuscript.

**Competing interests:** The authors have declared that no competing interests exist.

eukaryotes, SWR1, a SNF2 ATPase-associated chromatin remodelling complex, is responsible for correct incorporation of this histone variant. This study identified a SWR1-like complex in *T. brucei*. Depletion of the SNF2 ATPase resulted in a reduction of H2A.Z incorporation at the TSS and decreased steady-state mRNA levels accompanied by chromatin condensation. In addition to the SWR1-like complex, we also identified a trypanosome-specific HAT2 complex that includes the histone acetyltransferases HAT2, a key player in the H2A.Z incorporation process. This complex has a trypanosome-specific composition that is different from the NuA4/TIP60 complex in *Saccharomyces cerevisiae*.

## Introduction

Changes in chromatin structure can permit or restrict access to the DNA, thereby effectively regulating gene expression, replication, DNA repair and other nuclear processes in the cell [1]. This dynamic accessibility requires several molecular processes to make continuous remodelling possible and disturbance of these tightly-controlled mechanisms can cause severe damage. Aberrant gene expression, genomic instability due to inaccurate DNA repair, arrested DNA replication, and chromosomal translocation represent only a few examples caused by flawed chromatin structure [2–5].

Posttranslational modifications (PTMs) of histones play an essential role in chromatin accessibility. Histone acetylation, methylation and phosphorylation are linked to specific chromatin restructuring mechanisms [6–8]. These PTM-mediated processes can occur either by directly altering the strength of interactions between individual nucleosomes or by providing specific binding platforms for other protein complexes that alter chromatin structure [9, 10].

ATP-dependent chromatin remodelling enzymes belong to the switch 2/Sucrose non-fermentable 2 (SWI2/SNF2 or simply SNF2) superfamily. This superfamily was originally divided into four major subfamilies: 1) switch/sucrose non-fermentable (SWI/SNF), 2) chromodomain, helicase, DNA-binding (CHD), 3) inositol-requiring 80 (INO80) and 4) imitation switch (ISWI), but recent phylogenetic analyses facilitated a precise division into more subgroups [11, 12]. Proteins of all subfamilies contain a characteristic ATPase domain that is split into a DEAD-Box motif (DExx) and a helicase activity [11]. Complexes with these enzymes can vary in their composition from two subunits, such as the human RSF (remodelling and spacing factor) complex, to multimeric complexes like the *S. cerevisiae* INO80 complex with almost a dozen subunits [13, 14]. To enable interaction with PTMs, remodelling complexes often include variable proteins that exclusively bind to specific modifications. Chromodomain-containing proteins bind histone methylations, while bromodomains bind histone acetylations [10, 15].

Remodelling processes can be categorized by different modes of action: nucleosome translocation, nucleosome eviction and nucleosome remodelling. For example, the RSC complex can translocate an existing nucleosome several base pairs along the DNA to increase accessibility to DNA sequences [16]. Eviction of nucleosomes as well as nucleosome remodelling, in which canonical histones are replaced by their corresponding histone variants, rely often on an interplay of multiple proteins and protein complexes [17]. A well-studied example of such a histone exchange is the incorporation of the histone variant H2A.Z.

This histone variant is highly conserved among eukaryotes and only a few species (e.g. *Giardia and Trichomonas*) are known to lack an H2A.Z homologue [18, 19]. Nucleosomes containing H2A.Z play an important role in various cellular processes ranging from heterochromatin regulation and DNA repair to transcription regulation [20–23]. Correct incorporation of H2A.

Z into nucleosomes is mediated by the SWR1 (SWI2/SNF2 related 1) complex and requires well-regulated interactions between PTMs and protein complexes [10, 24–26]. For example, acetylation of histone H4 by the NuA4/TIP60 complex is essential for correct H2A.Z incorporation in *S. cerevisiae*. Acetylated H4 is bound by Bdf1 (bromodomain factor 1), which is part of the SWR1 complex [10, 27, 28]. In *S. cerevisiae* and humans, H2A.Z is also acetylated by the NuA4 (nucleosome acetyltransferase of H4)/TIP60 (HIV-1 TAT-interactive protein, 60kDa) complex [10, 29–33]. The cascade of histone H4 acetylation by an acetyltransferase complex followed by H2A.Z incorporation through SWR1 appears to be a highly conserved process in eukaryotes [34]. However, while yeast cells have the two distinct NuA4/TIP60 and SWR1 complexes for H4 acetylation and H2A.Z incorporation, the human p400/Tip60 and the Domino/dTIP60 hybrid complex in *Drosophila* combine both functions [21, 35]. In addition to p400/Tip60, humans possess a second acetyltransferase-independent complex, the SRCAP complex, which is also capable of incorporating H2A.Z [36, 37].

The function of H2A.Z in transcription regulation is context dependent. For example, it was shown that H2A.Z can block Notch and ΔNp63a target genes but promotes oestrogen receptor alpha dependent transcription [32, 38, 39]. PTMs also appear to define the function of H2A.Z-containing chromatin. Ubiquitination, SUMOylation as well as methylation and acetylation have been described for H2A.Z, linking its roles to X chromosome inactivation, DNA damage response, cell cycle progression and transcription regulation [32, 40–44]. Additionally, acetylation of H2A.Z was associated with active transcription in eukaryotes [32, 42, 45]. The removal of H2A.Z requires the INO80 complex, which contains the SNF2 ATPase, which belongs to the same subfamily as SWR1 [12, 46, 47]. Both complexes (INO80 and SWR1) share several protein components (reviewed in [48]). RuvB-like proteins, a YEATS (Yaf9, ENL, AF9, Taf14, and Sas5) domain-containing protein, actin, as well as actin-related proteins can be found in both complexes, with the latter two playing an important role in nucleosome binding [49, 50]. Only a few unique and non-conserved features distinguish the complexes from each other. For example, Znf-HIT1 (zinc-finger histidine triad1) domains-containing proteins such as Vps71 (vacuolar protein sorting 71 homologue)/SWC6 (SWR1 complex protein 6), specific bromodomain factors or the BCNT (bucentaur) /Cfdp1 (Craniofacial developmental protein 1) factor appear to be exclusively present in SWR1-like complexes. The yeast Ies2 (INO eighty subunit 2) /papa-1 (Pim-1-associated protein-1 (PAP-1)-associated protein-1) factor (INO80B in humans) has only been described in INO80-like complexes [51]. Despite a high degree of conservation between eukaryotic species, little is known about H2A.Z and its associated remodelling complexes in parasitic protists.

A connection between H2A.Z and transcriptional regulation was drawn in *Trypanosoma brucei*, the causative agent of African trypanosomiasis (sleeping sickness), very recently. It was shown that H2A.Z accumulates exclusively in the transcription start sites (TSS) of polycistronic transcription units (PTUs), a characteristic feature of genome organization in *T. brucei* [52, 53]. Another recent publication showed that both MYST-histone acetyltransferases HAT1 (histone acetyltransferase 1) and HAT2 are responsible for H2A.Z deposition at TSSs, although HAT1 appears to affect H2A.Z localization less than HAT2 [54]. Furthermore, it could be shown that acetylation of H4 and H2A.Z is mediated by two different histone acetyltransferases. HAT2 acetylates H4, while HAT1 acetylates H2A.Z. Depletion of HAT2 results in relocation of transcriptional initiation within the TSS [54]. Depletion of HAT1 affects RNA polymerase II (RNAP II) recruitment, which consequently reduces transcriptional activity in all PTUs [54]. However, neither a SWR1 nor INO80-like complex has been characterized in *T. brucei*.

Using co-immunoprecipitation (co-IP) experiments coupled to quantitative proteomics, we identified a multi-subunit protein complex with characteristics of a SWR1-like remodeller in

*T. brucei*. We additionally found that the histone acetyltransferases HAT1 and HAT2, which play a key role in H2A.Z-mediated transcription regulation, form two distinct complexes. Cell fractionation experiments and ChIP-seq analysis demonstrated that H2A.Z levels are significantly reduced at the TSSs after RNAi-mediated depletion of the SWR1-like subunit of this complex. Furthermore, luciferase reporter assays showed a reduced transcription activity within PTUs after RNAi, which was supported by Northern blot analysis showing a significant reduction of RNA transcribed by RNA polymerases I and II. By electron microscopy imaging we observed that depletion of the SWR1-like protein resulted in chromatin condensation similar to that observed after RNAP II or H2A.Z depletion. Our study therefore closes an important gap in understanding the process of H2A.Z-dependent transcription regulation in *T. brucei* by the identification and characterization of the corresponding chromatin remodelling complex.

## Results

### Identification of a novel SNF2 ATPase complex

To identify candidate SNF2-like proteins in *T. brucei*, we carried out a homology-based search in the TriTrypDB and identified 15 candidates that putatively belong to the SNF2 superfamily. In particular, we were interested in candidates with an insertion in the DEXH/Q motif between the DEAD-Box and helicase C motif, which has been shown to be characteristic for SNF2 proteins of the SWR-1-like subfamily [12] (S1A Fig). One putative candidate, Tb927.11.10730, matched this criterion. To test whether this candidate is a component of a SWR1-like remodeller complex, we wanted to carry out immunoprecipitations using a tagged version of the protein. Neither N- nor C-terminal tagging of Tb927.11.10730 proved possible, however. As an alternative strategy, we carried out co-immunoprecipitations (co-IPs) using the DNA helicase RuvB, which is an important and conserved part of both the INO80 and SWR1 complexes [51, 55]. Similar to other model organisms, two versions of the RuvB DNA helicase can be found in *T. brucei*, Tb927.4.1270 and Tb927.4.2000. In the following we will refer to these putative helicases as RuvB1 and RuvB2, respectively. Mass spectrometry (MS) analysis of the RuvB2 co-IP successfully identified the SNF2 candidate Tb927.11.10730 together with 14 additional proteins (Table 1 and S1B Fig) that were significantly enriched (p<0.01) and had high nuclear enrichment scores (NES) [56]. These proteins included known conserved components of the SWR1 remodeller such as YEATS domain-containing proteins and actin-like proteins.

To confirm these proteins as components of the complex and to identify factors that might have been missed in the RuvB2 co-IP, another round of co-IPs was performed. The proteins tagged for this reciprocal approach were Tb927.10.11690, a protein containing a YEATS domain (referred to as *Trypanosoma brucei* SWR1 complex protein 1 (hereafter *Tb*SWRC1), Tb927.11.5830, which contained an YL1 domain (hereafter *Tb*SWRC2) and the *Trypanosoma*-specific hypothetical protein Tb927.7.4040 (hereafter *Tb*SWRC4). *Tb*SWRC1 and *Tb*SWRC2 were promising candidates, as parts of the three-dimensional structure of *Tb*SWRC1 and *Tb*SWRC2 could be modelled with high confidence based on Yaf-9 (yeast AF9) and Vps72/SWC2, respectively, employing the Phyre2 homology modelling web tool [57]. In yeast, the YEATS domain of Yaf-9 is essential for histone binding and enables the accurate assembly of the SWR1 complex on the chromatin fibre [58, 59]. The conserved YL1 domain of Vps72 (vacuolar protein sorting 72 homologue)/ SWC2 (SWR1 complex protein 2) acts as a molecular lock to prevent H2A.Z eviction in the SWR1 complex and mediates, together with Arp5 proper nucleosome repositioning by the INO80 complex [55, 60]. The two proteins play a key role in both SWR1 and INO80 function and are conserved in SWR1 and INO80 complexes in several

**Table 1. Summary of the RuvB2 co-IP.** 15 proteins with a positive or unknown nuclear enrichment score were identified by MS in the RuvB2-HA co-IP. The nuclear enrichment score (NES; (52)) indicates the probability of a nuclear localisation based on cell fractionation combined with quantitative MS analysis. The "Annotation" column indicates the curated annotation that was found for the corresponding accession number on the TriTyp database. In the "p-value" column a probabilistic confidence measure (*P-value*) is assigned to each identified protein. The fold enrichment compared to the WT control is stated for every identified protein. The column "TrypTag localisation" indicates the cellular localisation of N- and C-terminally tagged proteins.

| Accession number | Annotation | fold enrichment | p-value | NES | TrypTag localisation |
|---|---|---|---|---|---|
| Tb927.4.2000 (RuvB2) | ruvB-like DNA helicase, putative | 8.10 | 1.50E-08 | 0.52 | nucleoplasm, cytoplasm, |
| Tb927.11.10730 (*Tb*SWR1) | SWI/SNF-related helicase, putative | 7.62 | 7.08E-08 | 5.04 | nucleoplasm, cytoplasm |
| Tb927.4.980 | Actin | 6.65 | 1.92E-05 | 5.71 | cytoplasm |
| Tb927.11.5830 (*Tb*SWRC2) | YL1 nuclear protein | 6.63 | 5.94E-07 | N/A | nucleoplasm, cytoplasm |
| Tb927.7.4040 (*Tb*SWRC4) | hypothetical protein | 6.62 | 1.56E-07 | 5.52 | nucleoplasm, cytoplasm |
| Tb927.4.1270 (*Tb*RuvB1) | ruvB-like DNA helicase, putative | 6.52 | 3.21E-05 | 0.69 | nucleoplasm, cytoplasm, |
| Tb927.10.11690 (*Tb*SWRC1) | YEATS family, putative | 6.24 | 1.70E-05 | 4.89 | nucleoplasm, cytoplasm |
| Tb927.10.2000 (*Tb*ARP3) | Actin like Protein | 6.11 | 1.11E-05 | 5.31 | cytoplasm, flagellar cytoplasm, nuclear lumen |
| Tb927.6.2570 (*Tb*ARP2) | putative SUMO-interacting motif containing protein | 5.80 | 6.16E-06 | 7.26 | nucleoplasm, cytoplasm |
| Tb927.8.600 (*Tb*SWRC5) | Bucentaur or craniofacial development, putative | 5.44 | 5.06E-07 | 2.52 | nucleoplasm, cytoplasm, mitochondrion |
| Tb927.3.3020 (ARP1) | Actin like Protein | 4.80 | 5.41E-04 | 3.43 | cytoplasm, nucleoplasm |
| Tb927.11.6290 (*Tb*SWRC6) | HIT zinc finger, putative | 4.70 | 1.02E-08 | 4.38 | nucleoplasm, cytoplasm |
| Tb927.11.16370 | SHQ1 protein, putative | 4.05 | 2.68E-07 | N/A | cytoplasm, flagellar cytoplasm, nucleoplasm, nuclear lumen |
| Tb927.9.5320 | nucleolar RNA binding protein | 3.63 | 6.18E-07 | 2.33 | nucleolus, nucleoplasm |
| Tb927.10.170 | pseudouridine synthase, Cbf5p | 1.90 | 9.47E-09 | 2.51 | nucleolus |

species [51]. TbSWRC4 was chosen as a third candidate as it appears to be exclusively found in *Trypanosoma brucei*, *Trypanosoma cruzi* and *Blechomonas ayalai*, thus representing a phylogenetically restricted complex member. Neither a BLAST search with the NCBI database, the EMBL InterPro database nor a homology modelling led to reliable information about a possible homologue of *Tb*SWRC4 in other organisms. This second set of co-IPs enriched 13 proteins (Table 2) as well as histones H2A, H2B, H3, H4 and the histone variant H2A.Z, which strongly suggested a role in nucleosome assembly.

In addition to the proteins that were identified in the initial RuvB2 co-IP, the reciprocal co-IPs could identify the conserved hypothetical protein Tb927.9.8510 (referred to as *Tb*SWRC3 in the following). Five proteins originally found in the RuvB2 co-IP were not enriched in the other three co-IPs, indicating they are more closely associated with RuvB2. Furthermore, Phyre2 modelling suggested that Tb927.6.2570, a potential SUMO-interacting motif-containing protein (referred to as *Tb*ARP2 in the following) is probably a third actin-like protein [57]. *Tb*SWRC3 was modelled as a potential SET (Su(var)3-9, enhancer-of-zeste and trithorax)-methyltransferase by Phyre2 [57]. A putative catalytic site specific to SET-methyltransferases could be annotated including three potential binding sites for the co-factor S-adenosylmethionine. The 13 proteins that were identified in at least three of the four co-IP experiments assemble the trypanosome SWR1-like complex (Table 2). The complex composition differs slightly

**Table 2. Characterisation of the novel SWR1 complex components.** 13 proteins with a positive or unknown NES were identified by MS analysis in at least three of the four co-IP experiments. The initial co-IP was performed with RuvB2 (Tb927.4.2000, the reciprocal co-IPs with the proteins Tb927.10.11690, Tb927.11.5830 and Tb927.7.4040 were performed to confirm the RuvB2 co-IP data). The "Annotation" column indicates the curated annotation that was found for the corresponding accession number in the TriTryp database. The "identified domains" column displays the domains that were found by BLAST search using the NCBI database. The Phyre2 modelling column indicates proteins that were identified by homology modelling. Coverage (Cov.) indicates the coverage in percent between query and template. The confidence (Conf.) represents the relative probability in percent (from 0 to 100) that the match between query and template is a true homology. The nuclear enrichment score (NES; (52)) indicates the probability of a nuclear localisation based on cell fractionation combined with quantitative MS analysis. The column "TrypTag localisation" indicates the cellular localisation of N- and C-terminally tagged proteins. The last column shows in which of the four co-IPs the protein could be identified.

| co-IPs | Gene ID | Annotation | Identified domains | Phyre2 modelling | NES | TrypTaglocalisation | Ident. in co-IP No. |
|---|---|---|---|---|---|---|---|
| No 1. | Tb927.4.2000 (RuvB2) | ruvB-like DNA helicase, putative | TIP-49 domain | ruvb-like protein 1/2 (Cov. 94% Conf. 100%) | 0.52 | nucleoplasm, cytoplasm, cytoplasm | 1–4 |
| No. 2 | Tb927.10.11690 (*Tb*SWRC1) | YEATS family, putative | YEATS-domain | Yaf-9 (Cov. 25% Conf. 98%) | 4.89 | nucleoplasm, cytoplasm | 1–4 |
| No. 3 | Tb927.11.5830 (*Tb*SWRC2) | YL1 nuclear protein, putative | YL1 | SWC2 (Cov. 24% Conf. 98%) | N/A | nucleoplasm, cytoplasm | 1, 3, 4 |
| No. 4 | Tb927.7.4040 (*Tb*SWRC4) | hypothetical protein | - | DAMP1 (Cov. 8% Conf. 98%) | 5.52 | nucleoplasm, cytoplasm | 1–4 |
| | Tb927.3.3020 (*Tb*ARP1) | Actin like Protein, putative | Actin | ARP5/8 (Cov.98% Conf. 100%) | 3.43 | cytoplasm, nucleoplasm | 1–4 |
| | Tb927.4.1270 (RuvB1) | ruvB-like DNA helicase, putative | TIP-49 domain | ruvb-like protein 1/2 (Cov. 99% Conf.100%) | 0.69 | nucleoplasm, cytoplasm, cytoplasm | 1–4 |
| | Tb927.4.980 | Actin | Actin | - | 5.71 | cytoplasm | 1–4 |
| | Tb927.6.2570 (*Tb*ARP2) | SUMO-interacting motif-containing protein | - | ARP8/9 (Cov. 79% Conf. 98%) | 7.26 | nucleoplasm, cytoplasm | 1–4 |
| | Tb927.8.600 (*Tb*SWRC5) | Bucentaur or craniofacial development, put. | BCNT-domain | - | 2.52 | nucleoplasm, cytoplasm, mitochondrion | 1–4 |
| | Tb927.9.8510 (*Tb*SWRC3) | hypothetical protein, conserved | - | Histone methyltransf.. SET7/9 Cov. 33% Conf. 96%) | N/A | cytoplasm, nucleolus, flagellar, cytoplasm, nuclear lumen, endocytic | 2, 3, 4 |
| | Tb927.10.2000 (*Tb*ARP3) | Actin like protein, putative | Actin | ARP5/8 (Cov.78% Conf. 100%) | 5.31 | cytoplasm, flagellar cytoplasm, nuclear lumen | 1–4 |
| | Tb927.11.10730 (*Tb*SWR1) | SWI/SNF-related helicase, putative | DEXQ-Box SRCAP; Helic. C | INO80/CHD1/SWR1/ RAD54 (Cov. 60% Conf. 100%) | 5.04 | nucleoplasm, cytoplasm | 1–4 |
| | Tb927.11.6290 (*Tb*SWRC6) | HIT zinc finger, putative | Zf-HIT1 | SWC6 (Cov. 27% Conf. 99%) | 4.38 | nucleoplasm, cytoplasm | 1–4 |

from the *Saccharomyces cerevisiae* SWR1 complex (Fig 1). Nevertheless, sharing of certain components such as a BCNT- and a Znf-HIT1-domain containing protein clearly hint towards the complex being a SWR1-like remodeller. Domains that are typical for an INO80 complex like the PAPA-1-domain or an HMG-Box are absent (Fig 2) [51]. However, further characterisation of the molecular functions of this novel complex and especially of the SNF2 ATPase were necessary to support a SWR1-like complex.

## Depletion of TbSWR1 leads to a reduction of chromatin-associated H2A.Z

Ultimately, as the INO80 complex represents the functional counterpart of the SWR1 complex and vice versa, RNAi-mediated depletion of the SNF2 subunit can reveal whether chromatin

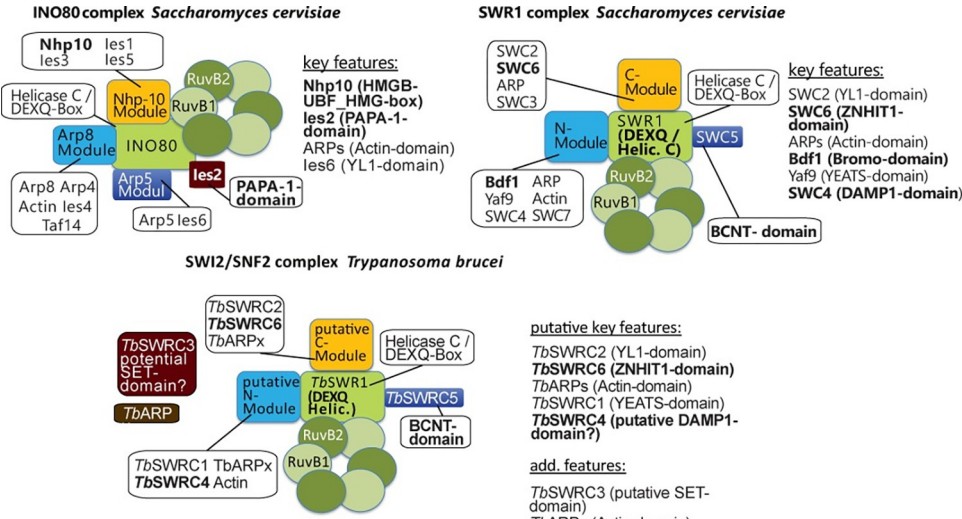

**Fig 1. The *Trypanosoma brucei* SNF2 ATPase complex exhibits characteristics of a SWR1 complex.** A comparison of the modular composition of the newly-identified SNF2 complex with the SWR1 and INO80 complexes of *S. cerevisiae*. **INO80 complex *S. cerevisiae*:** The Nhp-10 module forms a platform for nucleosome interaction. The Arp8 module is the nucleosome binding module, while the Arp5 module, which contains the YL1-domain protein Ies6 is responsible for the nucleosome remodelling step [14]. Ies2 with the PAPA-1-domain plays a structural role within the complex [14, 112]. INO80 specific domains are highlighted in bolt. **SWR1 complex *S. cerevisiae*:** Proteins listed under key features are essential for H2AZ incorporation [68]. The C-module of the SWR1 complex, which contains the Bromo-, YEATS- and SANT/DAMP1-domain mediates nucleosome affinity, while the N-module with the YL1- and Zinc finger ZNHIT1-domain is involved in the histone variant exchange reaction [49, 68]. SWR1 specific domains are highlighted in bolt letters [51, 68]. **SWI2/SNF2 complex *T. brucei*:** The identification of a BCNT- as well as a ZNHIT1 domain (highlighted in bold letters) hint toward the complex being a SWR1-like complex [51, 68]. Structure of the SWI2/SNF2 complex and its modules is only putative. The potential interaction interface of the species-specific proteins *Tb*SWRC3 and *Tb*ARPx is unknown.

association of H2A.Z will increase, indicating an INO80-like function, or if H2A.Z incorporation will decrease, implying an SWR1 functionality. We thus depleted the SNF2 ATPase (Tb927.11.10730) by tetracycline-inducible RNAi (Fig 2). The depletion resulted in a severe growth phenotype with an almost complete growth arrest 24h post-induction (Fig 2A). FACS analysis of propidium iodide-stained cells showed almost unchanged viability of the population 24h after induction, and approximately 70% living cells after 48h, however (Fig 2B). Cell fractionation experiments were performed to determine whether the amount of nucleosome-associated H2A.Z decreased in comparison to core histones following depletion of the SNF2 ATPase (Fig 2C and 2D). Proteins in the insoluble chromatin fraction were analysed by Western blot with antibodies specific for histone H3 and the variant H2A.Z. While levels of H2A.Z and a Ty1-tagged version of H2A.Z were significantly reduced over time, H3 levels appeared not to be affected, indicating that this SNF2 ATPase is indeed associated with efficient incorporation of H2A.Z into chromatin (Fig 2C and 2D). A second clone (*Tb*SWR1 c4) produced a similar albeit less severe phenotype (S1C Fig). Cell cycle analysis of *Tb*SWR1-depleted cells showed a reduction of G1 and to a lesser extent G2 cells during the time course and a continued accumulation of anucleate parasites (S1D Fig). As both the YEATS domain and the YL1 domain in the SWR1 complex play a key role in H2A.Z incorporation [49], we also investigated how the loss of *Tb*SWRC1 and *Tb*SWRC2 affected chromatin-associated H2A.Z levels. Though SNF2 ATPase depletion resulted in stronger effects, the reduction of chromatin-associated H2A.Z was also observable upon depletion of the YEATS domain-containing protein *Tb*SWRC1 and YL1 domain protein *Tb*SWRC2 (S1D Fig and S1E Fig), emphasizing the

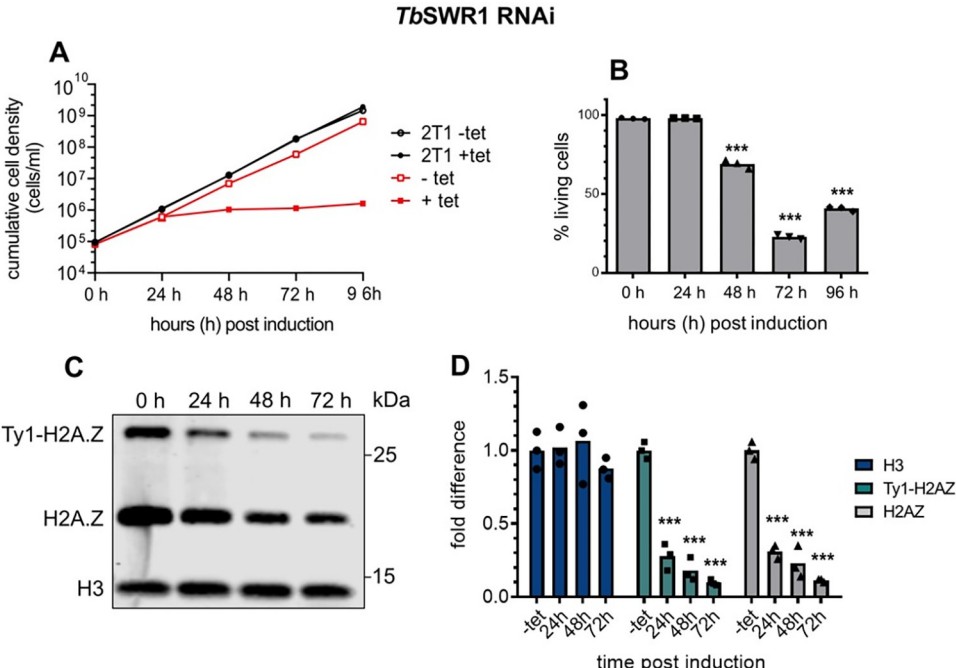

**Fig 2. Loss of *Tb*SWR1 leads to cell death and a reduction of chromatin-associated H2A.Z. (A)** Growth of parasites was monitored for 96 hours after RNAi-mediated depletion of *Tb*SWR1 (Tb927.11.10730) using tetracycline (tet). The parental 2T1 cell line was used as a control (n = 3). **(B)** Quantification of live/dead staining with propidium iodide of *Tb*SWR1-depleted cells at the indicated timepoints post-induction. Analysis was done by flow cytometry (n = 3). **(C)** Western blot analysis of the insoluble nuclear fraction with antibodies specific for histone H3 and the histone variant H2A.Z. Lysates from an equal number of cells ($2 \times 10^6$ per lane) were analysed for each timepoint. **(D)** Quantification of chromatin-associated H3 (dark blue), Ty1-H2A.Z (turquoise) and H2AZ (grey) (N = 3 for all depicted experiments; *** = p-value <0.001; ** = p-value 0.001–0.01; * = p-value 0.01–0.05).

contribution of the additional complex members in H2A.Z incorporation. As fractionation experiments showed a significant reduction of chromatin-associated H2A.Z, we were confident that this novel SNF2 ATPase is indeed a SWR1 homologue in trypanosomes. To analyse the effect of *Tb*SWR1 depletion more precisely, a ChIP-seq assay using a cell line with a Ty1-tagged H2A.Z allele was performed. Previous studies showed that Ty1-tagged H2A.Z is efficiently incorporated into nucleosomes at TSSs and our fractionation experiments showed that Ty1-H2A.Z is chromatin-associated (Figs 2 and S1C Fig). To avoid strong secondary effects, we chose *Tb*SWR1 c4, which had a milder phenotype, and an early time point after RNAi induction (S1C Fig). In line with our fractionation experiment, we could demonstrate a reduction of Ty1-H2A.Z at TSSs after 48 h post induction of *Tb*SWR1 depletion (Fig 3) clearly establishing the SWR1-like functionality.

## Depletion of HAT2-associated Bdf3 reduces chromatin-associated H2A.Z

SWR1 regulation and H2A.Z acetylation by the NuA4 complex has been extensively described in yeast [10, 29, 30, 61]. Recent studies in *T. brucei* linked acetylation of H4 and H2A.Z. with the two histone acetyltransferases HAT2 and HAT1 (S1G Fig). Since H2A.Z incorporation into nucleosomes was linked to HAT2 activity [54] we decided to investigate the complex composition of HAT2. Co-IP experiments with HA-tagged HAT2 identified a complex that consists of Bdf3, HAT2 and several other proteins including factors with an ENT (*EMSY* N-terminal)-domain and an FHA (fork head-associated)-domain (S1A Table and S1J Fig). Bdf3

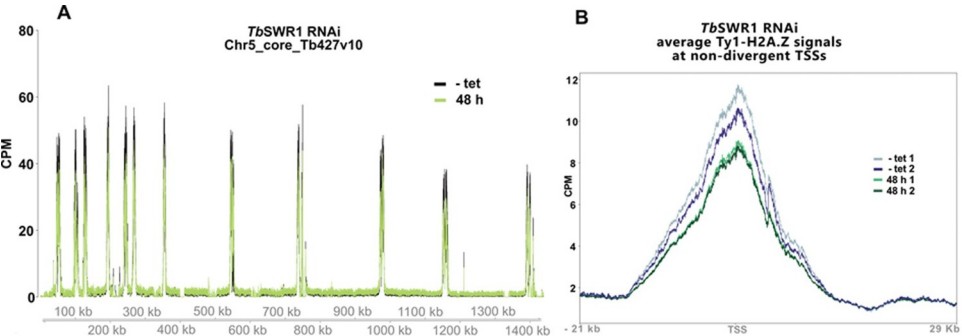

**Fig 3. Loss of *Tb*SWR1 (Tb927.11.10730) leads to a reduced H2A.Z deposition at TSS. (A)** ChIP-Seq analysis of distribution of Ty1-tagged H2A.Z before (black) and after (green) RNAi-mediated depletion of *Tb*SWR1 (48h post induction) revealed a reduction of chromatin associated Ty1-H2A.Z. Depicted is a representative region of chromosome 5. Data (n = 2) were normalised to the total number of reads and plotted as counts per million reads (CPM). **(B)** Average Ty1-H2A.Z signal across non-divergent TSSs. Peaks derived from two non-tetracycline induced reference samples depicted in blue and black. Peaks derived from two samples after 48 h of RNAi-mediated depletion of *Tb*SWR1 depicted in green and dark-green.

was particularly interesting because these factors play a key role in linking H4 acetylation to SWR1 activity and it has been associated with H2A.Z in trypanosomes already. ChIP-seq experiments with Bdf3 revealed a co-localization with histone H4K10 acetylation and H2A.Z in TSSs of *T. brucei* [52]. We performed a reciprocal Bdf3 co-IP and could confirm the HAT2 complex composition in trypanosomes with 10 out of 11 proteins reproducibly enriched (S1A Table and S1J Fig). Furthermore, RNAi-mediated depletion of Bdf3 caused a severe growth defect similar to the phenotype observed after *Tb*SWR1 depletion. Importantly, the amount of chromatin-associated H2A.Z significantly decreased after Bdf3 depletion (Fig 4), suggesting that it is also involved in H2A.Z deposition.

## Loss of *Tb*SWR1 leads to reduced RNA levels

The contribution of H2A.Z to transcription regulation has been extensively described in various organisms and a recent study from the Siegel laboratory showed that H2A.Z acetylation by HAT1 is required for RNAP II transcription regulation in *T. brucei* [54, 62] and that loss of H2A.Z acetylation affects the transcription of PTUs [54]. To determine the effect of SWR1 complex components on transcription of PTUs, we inserted a luciferase reporter construct into the tubulin array of several RNAi cell lines. Luciferase reporter assays have been used before in *T. brucei* to study transcriptional activity [63]. A significant decrease of luciferase activity following *Tb*SWR1 depletion could already be observed after 24 h. Luciferase activity decreased to approximately 60% and 30% after 36 h and 48 h induction of RNAi, respectively (Fig 5A). Cell death did not contribute substantially to decrease of luciferase signals because approximately 90% and more than 60% of the parasites were still alive after 36 and 48 hours (Fig 5B), respectively, suggesting that that the loss of luciferase activity is directly caused by the depletion of *Tb*SWR1. Depletion of H2A.Z itself showed comparable results but with faster kinetics (Fig 6). Although 96% of the cells were alive 24 h post-induction, a reduction of the luciferase to approximately 50% compared to un-induced parasites could be observed. For comparison and as a positive control, the loss of luciferase activity was also investigated in RNAP II-depleted cells. A loss of activity of 85% after 24 h of RNAi induction could be observed while approximately 85% of the cells were still alive (Fig 5D and 5E). In line with the two acetyltransferases HAT1 and HAT2 being linked to transcription and H2A.Z regulation, a similar reduction of mRNA levels was observed. HAT1 depletion caused a reduction to 56%

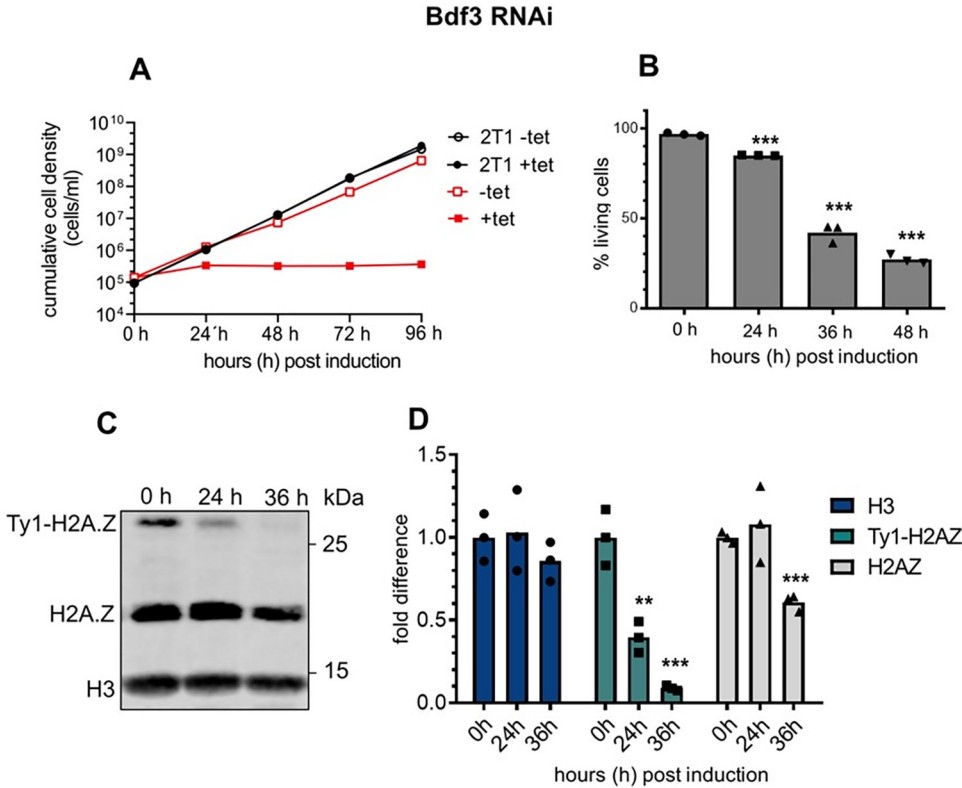

**Fig 4. Loss of Bdf3 leads to cell death and a reduction of chromatin-associated H2A.Z. (A)** Growth of parasites was monitored for 96 hours after RNAi-mediated depletion of Bdf3 using tetracycline (tet) induction. The parental 2T1 cell line was used as a control (n = 3). **(B)** Quantification of live/dead staining with propidium iodide of Bdf3-depleted cells. Analysis was done by flow cytometry (n = 3). **(C)** Western blot analysis of the insoluble nuclear fraction with antibodies specific for histone H3 and the histone variant H2A.Z. Lysates from an equal number of cells ($2x10^6$ per lane) were analysed for each timepoint. **(D)** Quantification of chromatin-associated H3 (dark blue), Ty1-H2A.Z (turquoise) and H2AZ (grey); (N = 3 for all depicted experiments; *** = p-value <0.001; ** = p-value 0.001–0.01; * = p-value 0.01–0.05).

and 71% in two independent clones 48 h post induction (S1H Fig). We could also detect a loss of luciferase activity to approximately 70% and 80% in HAT2-depleted cells compared to unin-duced cells (S1I Fig). In both experiments, we observed only a mild growth defect and a pro-portion of dead cells below 10% indicating that the loss of activity is not caused by dead cells (S1H Fig and 1I Fig).

Since these luciferase experiments provide only indirect information about polymerase activity, we next assessed steady-state mRNA levels directly in two independent *Tb*SWR1 and one RNAP II RNAi cell lines (Fig 7). Total RNA was extracted from *Tb*SWR1- and RNAPII-depleted cells for Northern blot analysis. A clear reduction of total mRNA and ribosomal RNA could be detected 48 hours after *Tb*SWR1 depletion. Analysis of RNA derived from RNAP II-depleted cells showed similar results but as observed before with faster kinetics. These North-ern blot analyses confirmed the initial observations demonstrating that mRNA and rRNA lev-els decreased approximal 2-fold 48 h after RNAi induction, with a stronger effect for *Tb*SWR1 c6 compared to *Tb*SWR1 c4 (Fig 7), in agreement with the reduction of H2A.Z in chromatin (Fig 2). Our data indicate that *Tb*SWR1 has a conserved role in transcription regulation which is typical for all SWR1-like proteins described so far.

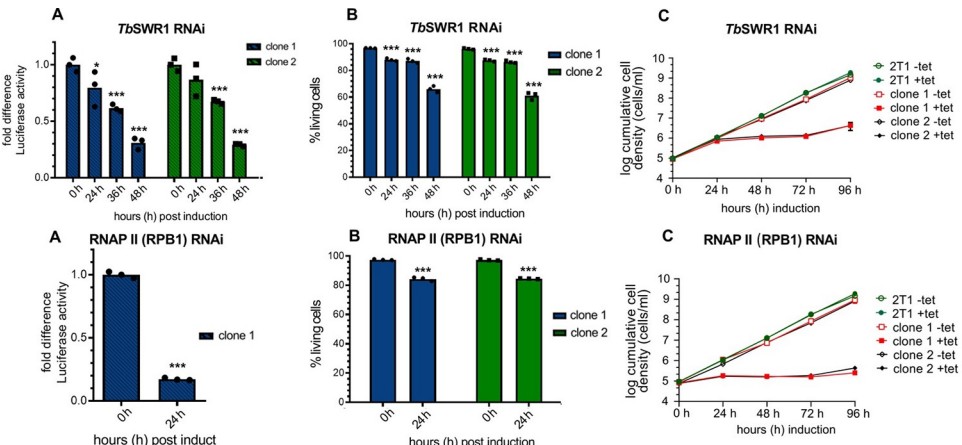

**Fig 5. Depletion of TbSWR1 caused a decrease of reporter luciferase activity within a PTU.** A luciferase reporter construct was integrated into the tubulin array in both RNAi cell lines. The *Tb*SWR1 RNAi clone 6 was used for transfection (Figs 3 and S1C Fig). Samples for the luciferase assay were normalised to cell numbers. **(A)** Luciferase activity was monitored for 48 h after induction of RNAi in two independent clones. Values of non-induced cells were set to 1. **(B+E)** Live/dead staining of each RNAi cell line was performed in triplicates at the same time points. **(C+F)** Growth of parasites was monitored for 96 hours after RNAi-mediated depletion of TbSWR1 and RBP2 using tetracycline (tet) induction. The parental 2T1 cell line was used as a control (n = 3). **(D)** As a positive control, luciferase activity of the same reporter construct was measured in an RNAP II RNAi cell line. (N = 3 for all depicted experiments; *** = p-value <0.001; ** = p-value 0.001–0.01; * = p-value 0.01–0.05).

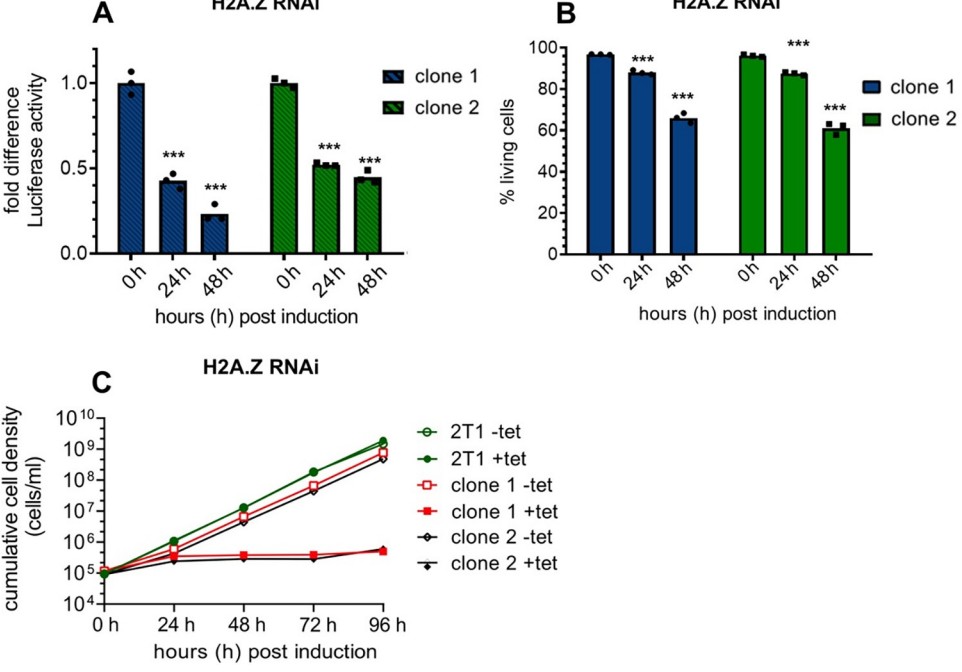

**Fig 6. Depletion of the histone variant H2A.Z leads to a decreased luciferase activity within a PTU.** A luciferase reporter construct was integrated into the tubulin array of a H2A.Z (Tb927.7.6360) RNAi cell line. Samples for the luciferase assay were normalised to cell numbers. **(A)** Luciferase activity was monitored for 48 h after induction of RNAi in two independent clones. Values of non-induced cells were set to 1; (N = 3); **(B)** Live/dead staining of each RNAi cell line was performed in triplicates at the same time points. **(C)** Growth of parasites was monitored for 96 hours (N = 3) after RNAi-mediated depletion of H2A.Z using tetracycline (tet) induction. Growth of the parental 2T1 cell line was measured for 96h as a control. (N = 3 for all depicted experiments; *** = p-value <0.001; ** = p-value 0.001–0.01; * = p-value 0.01–0.05).

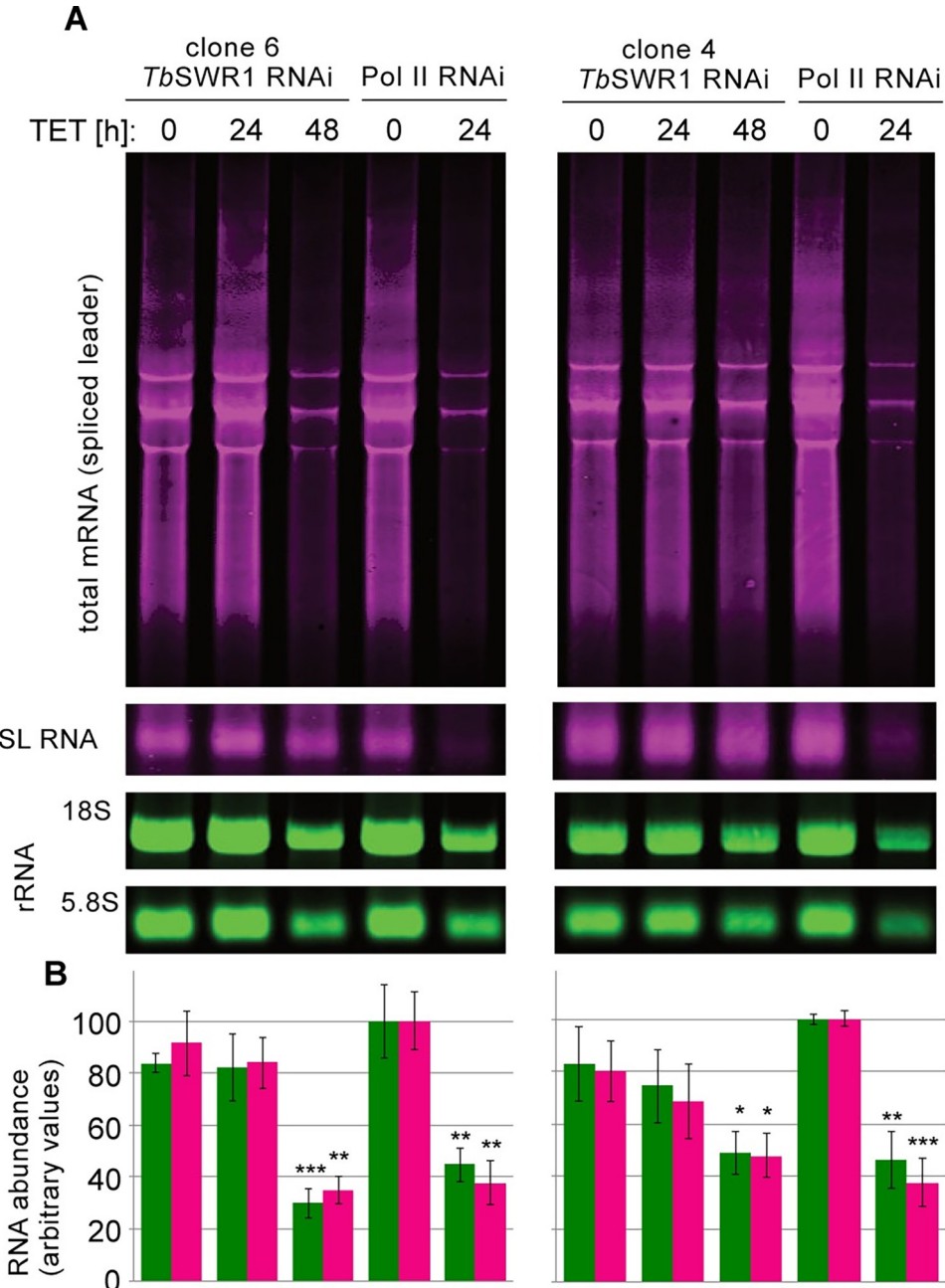

**Fig 7. Loss of *Tb*SWR1 leads to a reduction of mRNA and rRNA. (A)** Representative Northern blot of two *Tb*SWR1-depleted cell lines. RNAP II-depleted cells were used as a control. The samples were normalised to cell numbers. Upper and middle panel were hybridised with a probe specific for the spliced leader RNA (mini exon, red) the two lower panels with probes specific for ribosomal RNA (rRNA, green) **(B)** Quantification of results from three replicates showing total mRNA levels in red and 5.8S rRNA in green. Values of un-induced RNAP II RNAi cell lines were set to 100%.

## Depletion of *TbSWR1* affects chromatin structure

In other organisms, cessation of transcriptional activity causes massive changes in chromatin structure and nuclear architecture [64, 65]. Electron microscopy images of *Tb*SWR1-depleted cells showed that after 24 h of RNAi-mediated depletion of *Tb*SWR1 an increased

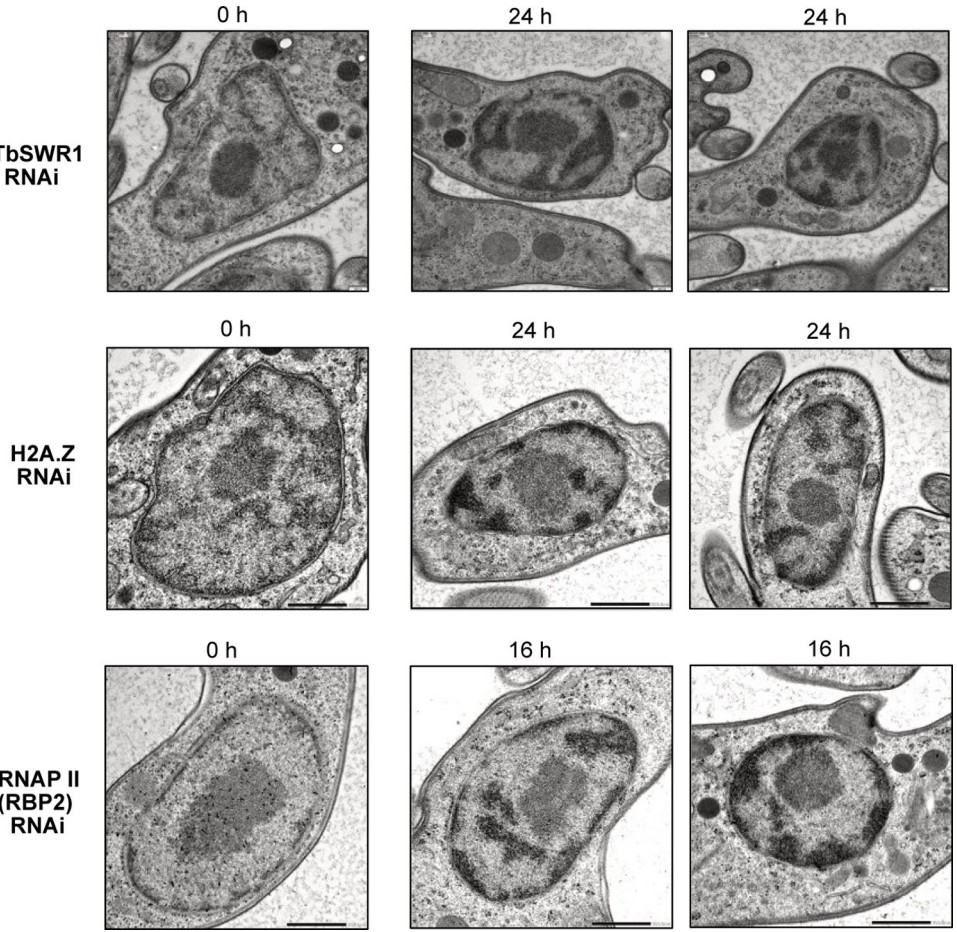

**Fig 8. Loss of *Tb*SWR1 and RNAP II leads to chromatin condensation.** Representative electron microscopy images of the nucleus of *Tb*SWR1-, H2A.Z- and RNAP II depleted and non-depleted parasites as indicated. Depletion of the proteins resulted in large black patches of condensed chromatin. Scale bar, 200 nm for *Tb*SWR1 RNAi and H2A.Z RNAi images, 500 for RNAP II (RBP2) RNAi images. Images of *Tb*SWR1 RNAi cells were obtained using a TEM (transmission electron microscope), images of RNAP II RNAi cells with a STEM (scanning transmission electron microscope).

condensation of chromatin in the nucleus in comparison to non-induced cells could be observed (Figs 8 and S1L Fig). Such a condensation could be seen in several albeit not all cells. A knockdown of the RNAP II and H2A.Z led to a comparable phenotype but with different kinetics. While RNAP II-depleted cells showed a condensation 16 h post induction, H2A.Z depleted cells exhibited chromatin condensation after 24 h of RNAi induction (Figs 8 and S1L Fig). Overall, the temporal behaviour of chromatin condensation seems similar in *Tb*SWR1- and H2A.Z-depleted cells, supporting the function of *Tb*SWR1 as a histone remodelling complex important for H2A.Z deposition.

In summary, purification, molecular characterization and RNAi-mediated depletion of critical components of a novel SWR1 chromatin remodeller complex could shed some light on the so far insufficiently investigated mechanism of transcriptional regulation in African trypanosomes.

## Discussion

Incorporation of H2A.Z by a SWR1 remodelling complex is a conserved mechanism of transcriptional regulation in many eukaryotic cells [10, 22, 23, 32, 66]. The goal of our study was to

screen for an SWR1-like complex in the unicellular parasite *T. brucei* with a combination of quantitative mass spectrometry and co-IP experiments. The trypanosome SWR1 complex so identified contains an SNF2-ATPase, which has characteristics of other members of the SWR1 subfamily [12]. In the process of this work, we also described the trypanosome HAT2 complex bearing a unique complex composition. While very recent work from Staneva et al. already indicated the presence of an SWR1-like and an HAT2 complex [67], we here functionally characterize the two complexes and show that both are associated with H2A.Z incorporation in *T. brucei*. Additionally, we demonstrated that the loss of the SNF2 ATPase *Tb*SWR1 resulted in decreased incorporation of H2A.Z, a loss of RNAP I- and RNAP II-dependent transcripts, and a condensation of chromatin.

## Composition of the *Tb*SWR1 and HAT2 complexes

Co-IP experiments with 4 different subunits combined with quantitative proteomics provided a clear picture and was supplemented with a thorough *in silico* analysis of the identified proteins, with particular reference to homology with the SWR1 complex from *S. cerevisiae*. We identified homologues for most of the essential SWR1 complex components (reviewed in [68]), such as an SNF2 family helicase (*Tb*SWR1), actin-related proteins (*Tb*ARP1-3), a YEATS domain containing protein (*Tb*SWRC1), and a YL1-domain containing protein (*Tb*SWRC2). The BCNT-domain and the Znf-HIT1-domain that are characteristic for a SWR1 complex [51] could be identified in *Tb*SWRC5 and *Tb*SWRC6 (Fig 2) as well. We also found that the composition of the SWR1 complex in trypanosomes has some distinct differences. The presence of a DNA methyltransferase 1-associated protein 1 (DAMP1)-domain [51] could not be verified in the *Tb*SWR1 complex so far. A fragment of a DAMP-1-domain was annotated in *Tb*SWRC4 by Phyre2 homology modelling but only with an insufficient confidence. To our surprise, a bromodomain protein—typically part of SWR complexes in other species—was missing from our co-IP data. In *S. cerevisiae*, Bdf1 is able to bind acetylated histone H4 [69] and is part of the SWR1 complex. While it is possible that the absence of a bromodomain factor is due to different interaction affinities and it was lost during purification, our data suggest that the core *Tb*SWR1 complex may not contain such a bromodomain protein. *Tb*SWRC1, which has a YEATS domain and was shown to effectively bind to acetylated histone H3 [59], may bind acetylated histone H4 instead. Besides the absence of a Bdf homologue in the complex, the identification of a third actin-related protein (S1B Fig) and a potential SET domain-containing protein (named *Tb*SWRC3; S1B Fig) was unexpected. SET methyltransferases play important roles in transcription regulation (reviewed in [70]), suggesting that *Tb*SWRC3 represents an important part of the transcriptional regulatory machinery in *T. brucei*. Depending on the substrate of *Tb*SWRC3, the additional Arp protein might be necessary to create a larger nucleosome interaction interface or add a specific functionality. The SWR1 complex and H2A.Z have been shown to be involved in many cellular processes including DNA repair [71–73]. As part of the NuA4 complex in *S. cerevisiae*, Arp4 plays an important role in DNA damage repair by recognizing γH2A [74]. Besides a putative role in correct *Tb*SWRC3 positioning, the additional Arp protein in the *Tb*SWR1 complex might be a γH2A recognition module. Given that H2A.Z was shown to be actively incorporated at DNA double strand-breaks [71], the *Tb*SWR1 complex might be directly recruited to sites of DNA damage. As *T. brucei* lacks non-homologous end joining (NHEJ), H2A.Z might be required to mark areas around DSBs as nucleosome free regions to expose the DNA for the repair process by homologous recombination or micro-homology end joining [75].

In course of investigations that focused on Tb927.9.2910, a protein annotated as a NuA4 subunit, we were able to identify a potential protein complex assembled around HAT1 and

HAT3. Co-IP experiments with Tb927.9.2910 identified three proteins that could be linked to the NuA4 complex after Phyre2 modelling: Tb927.1.650 (an Eaf3/MORF4 homologue), Tb927.7.5310 (a Yaf9/GAS41 homologue) and Tb927.10.14190 (a homologue of Epl1). A reciprocal co-IP with the Eaf3/MORF4 homologue could also precipitate HAT1 (but not HAT3) and the three proteins homologous to NuA4 subunits (S1B Table). These data indicate the presence of at least one HAT complex with similarities to a NuA4 complex in *T. brucei* (S1K Fig).

In contrast, the *T. brucei* HAT2 complex composition appears to be significantly different compared to the NuA4/TIP60 complex in *S. cerevisiae* [76]. We could show that Bdf3, a factor that has been shown to locate at the TSSs [77] is involved in H2A.Z incorporation as a part of the HAT2 complex (Figs 4 and S1J Fig), but the function of the other novel complex components remain elusive. We identified two hypothetical proteins without conserved domains and three proteins, which either possess an ENT domain, an FHA domain, or a BTB domain (S1A Table). BTB domains and the ENT domain are associated with chromatin structure regulation [78–80]. None of trypanosome factors appear to have homologues in the NuA4/TIP60 complex [76]. FHA domains are known for recognition of phosphorylate threonine. Phospho-threonine can be found on a broad range of proteins, including kinases, transcription factors and RNA-binding proteins [81, 82]. The presence of an FHA-domain could hint towards phosphorylation events that are important for H2A.Z incorporation. Based on our data, we cannot assess the nature of a potential molecular interaction between HAT2 and the *Tb*SWR1 complex. Interestingly, proximity labelling assays with *Tb*SWRC2 showed that both complexes are in close proximity to each other. Given that the FHA and EMSY domain could be linked with DNA damage repair and chromatin remodelling, it will be interesting to analyse the HAT2 complex structure and the function of its different subunits to learn more about its role in H2A.Z incorporation and other nuclear processes.

## Transcription regulation by the *Tb*SWR1 complex

H2A.Z is involved in transcription regulation in many organisms including *T. brucei* [22, 32, 54]. Northern blot analysis confirmed a loss of RNAPI- and RNAPII-dependent transcripts after the downregulation of *Tb*SWR1 (Fig 7), presumably due to the decrease in H2A.Z incorporation. Consequently, HAT1 loses its substrate and therefore cannot initiate transcription [54]. The observed downregulation of total mRNA and rRNA is not seen for the spliced leader mRNA, which appears to be mainly unaffected by *Tb*SWR1 depletion (Fig 7). This is in agreement with a promotor-dependent and mechanistically different transcription of the spliced leader compared to that of PTUs [83]. The loss of rRNA was surprising, but since a similar loss was observed after RNAP II downregulation (Fig 7), it is likely due to secondary effects. Many genes of RNAP I subunits are encoded in PTUs, which are transcribed by RNAP II. A lack of RNAP II activity might therefore result in a reduced amount of RNAP I transcripts, which finally leads to a reduced amount of rRNA.

## Changes in chromatin structure after loss of H2A.Z and RNAP II

The chromatin condensation we observed (Fig 8) might be a direct consequence of the loss of RNAP II activity. It has been described previously that condensation of chromatin can be induced in mouse and human cells by treatment with alpha-amanitin, an RNAP II and RNAP III inhibitor [64, 65, 84]. Alpha-amanitin treatment also blocks RNAP II and III transcription in trypanosomes [85]. Surprisingly, it also affected genes that were under the regulation of an exogenous T3 promoter indicating that open chromatin structure is dependent on transcription activity [85]. Given that a knockdown of *Tb*SWR1, RNAP II and H2A.Z led to a reduction

of luciferase activity and mRNA abundance (Figs 5, 6 and 7), it is likely that the chromatin condensates we observed is a consequence of the RNAP II transcription shutdown.

The identification and subsequent characterization of a SWR1-like complex in *T. brucei* closes the gap in transcription regulation between HAT2 and HAT1, because we identified the remodeller responsible for H2A.Z incorporation.

Interestingly, a protein complex such as INO80, which mediates the removal of histone variants still remains elusive in *T brucei*. Despite the fact that RuvB1 and RuvB2 are shared subunits of the INO80 and the SWR1 complex in other organisms [68], there is no hint of an INO80 complex in our co-IP data. Given the high degree of conservation of the two SNF2 ATPase complexes between different species, a RuvB co-IP should have identified at least a second SNF2-like protein. In the absence of an INO80 complex and with trypanosome-specific subunits the *Tb*SWR1 complex could have a dual function, also facilitating H2A.Z removal. Alternatively, locus-specific removal of H2A.Z might not be necessary considering the constitutive transcription initiation in *T. brucei*.

In addition to TbSWR1 only one additional chromatin remodelling complex, the ISWI complex, has so far been identified in *T. brucei* [86]. As we have shown here, epigenetic complexes in trypanosomes may have evolved differently compared to yeast and humans. It is thus crucial to identify other chromatin-remodelling complexes to understand fundamental biological processes in trypanosomes such as DNA repair and recombination, chromatin dynamics during developmental differentiation or monoallelic transcription of surface proteins.

## Methods

### Trypanosoma brucei culture

Monomorphic *Trypanosoma brucei* Lister 427 bloodstream form (BSF) MiTat 1.2 (clone 221a) strain, and a derivative '2T1' strain [87] that contains a puromycin-tagged ribosomal spacer for directed integration of the RNAi construct and expresses a Tet repressor protein, were cultivated in HMI-9 medium [88] with 10% heat-inactivated fetal calf serum (FCS; 56˚C for 1 h) at 37˚C and 5% CO2. Strains were cultured with their attendant drug selection with the following concentrations used: 2.5 μg/ml G418 (neomycin), 5 μg/ml hygromycin, 0.1 μg/ml puromycin, 5 μg/ml blasticidin, 2.5 μg/ml phleomycin. RNAi was induced with 1 μg/ml tetracycline. Reduction of mRNA was monitored by RNA/FISH (S1M Fig). Growth rates were monitored for 96 h and cell densities were determined every 24 h using a Coulter Counter Z2 particle counter (Beckman Coulter). Procyclic forms (PCF; strain 427) were cultured in modified SDM-79 with 10% heat-inactivated FCS (Sigma) at 27˚C. BSF and PCF parasites were transfected as previously described [89], with independent clones obtained by limiting dilution.

### MNase-ChIP sequencing

The MNase-ChIP experiments were performed with 2T1 cells and the 2T1 TbSWR1 RNAi cell line with a single Ty1-tagged H2A.Z allele. Functionality of the Ty1-tagged H2A.Z has been described in [54]. In brief, $2 \times 10^8$ cells were harvested, crosslinked in 1% formaldehyde, and subsequent lysed using 200 μM digitonin (final concentration). Chromatin was fragmented by incubating the cells with 4 U $μl^{-1}$ MNase (Sigma-Aldrich) for 10 min at 25˚C. For a detailed ChIP protocol see ref. [90]. Immunoprecipitation was performed using Dynabeads M-280 sheep anti-mouse coupled to 10 μg monoclonal, purified anti-Ty1 (BB2) mouse antibody [91], overnight (~16 h) at 4˚C in the presence of 0.05% (final concentration) sodium dodecyl sulfate (SDS). Immunoprecipitated material was washed with RIPA-Buffer (50 mM HEPES-KOH (pH 7.5), 500 mM LiCl, 1 mM EDTA, 1% (vol/vol) IGEPAL CA-630, 0.7% (wt/vol) Na-Deoxycholate. LiCl and Na-Deoxycholate dissolved separately in, mix and add remaining

components. Store at 4°C.) and eluted with 200μl elution buffer (50 mM Tris-HCl (pH 8.0), 10 mM EDTA, 1% (wt/vol) SDS) at 65°C for 30 min. Cross-links were reversed at 65°C for ~16 h in the presence of 300 mM NaCl (final concentration). 4 μl of 10 mg/ml RNaseA were added to the ChIP sample and incubated at 37°C for 2 h. 4 μl of 10 mg/ml proteinase K were added to the ChIP sample and incubated at 55 C for 2 h in a heat block. The tubes were centrifuged at 10,000 x g for 10 min at RT. DNA was purified with the Macherey & Nagel NucleoSpin Gel and PCR Clean-up kit (the NTB buffer was used instead of the NTI buffer due to the high SDS concentration within the samples). Sample purification was performed according to the manufacturer's instructions. The ChIP sample was eluted with 16 μl and the input sample with 26 μl of NE buffer. The DNA library preparation was performed using NuGEN´s Ovation Ultralow System V2 (M01379 v5). Libraries were prepared with a starting amount 7ng of DNA and were amplified in 8 PCR cycles. Libraries were profiled with a 2100 Bioanalyzer (Agilent technologies) and quantified using the Qubit dsDNA HS Assay Kit, in a Qubit 2.0 Fluorometer (Life technologies) set to high sensitivity. All 14 samples were pooled in equimolar ratio and sequenced on a NextSeq500 High output Kit, PE for 2x 42 cycles plus 8 cycles for the index read. The ChIPseq metadata is available at https://www.ncbi.nlm.nih.gov/bioproject/PRJNA744383

## Bioinformatics analysis

**Reads processing and mapping.**   Library quality was assessed with FastQC version 0.11.8 before being aligned against the *T. Brucei* genome assembly TriTrypDB-48_TbruceiLister427_2018_Genome.fasta and the TriTrypDB-48_TbruceiLister427_2018.gff annotation file [92, 93]. Such alignment was performed with bowtie2 aligner [94] version 2.3.4 (options:—very-sensitive—phred33—fr—maxins 1000—minins 0—end-to-end). Multimapping reads were filtered out and the resulting unique reads were sorted and indexed using SAMtools [95]. Peaks mapping to annotated features in the.GFF file were quantified with MACS2 [96] version 2.1.2 (options:—g 35000000—bw 150—min-length 150—format BAMPE—keep-dup—auto) with an FDR cutoff of 5%. Chromosome coverage tracks were generated with deepTools [97] version 3.1 (bamCoverage, options:—binSize 10—normalizeUsing CPM) and plotted using Gviz [98] in an R framework [99]. Finally, signal at TSS was quantified (computeMatrix reference-point, options:—referencePoint TSS -b 25000 -a 25000—skipZeros) and plotted (plotProfile, options:—perGroup) using deepTools and a custom.BED file containing transcript annotations at the peaks.

## Differential binding analysis

Further filtering and an exploratory analysis was performed in an R framework including ggplot2 [100]. Overall experimental quality was assessed via IP strength [101]. Differential binding comparisons were performed with DiffBind package [102] and differential peaks were selected with a 5% FDR. Finally, differential peaks were functionally annotated with ChIPseeker package [103] were a -3000/ + 3000 region around the TSS was considered as promoter.

## Extraction of chromatin-associated proteins

The amount of incorporated H2A.Z was determined by Western blot analysis after fractionation of the cells. H3 was used as a control. We followed the extraction protocol described by Kraus *et al.* [54]. Analysis of chromatin-associated proteins was performed in 2T1 cells with a tagged Ty1-H2A.Z allele. Cells were harvested by centrifugation (4°C, 1500 × *g* for 10 min) and washed in 1 ml of trypanosome dilution buffer (TDB; 5 mM KCl, 80 mM NaCl, 1 mM MgSO$_4$, 20 mM Na$_2$HPO$_4$, 2 mM NaH$_2$PO$_4$, 20 mM glucose, pH 7.4) followed by an additional

centrifugation (4˚C, 1500 × *g* for 10 min). The cell pellet was solubilized in CSK-buffer (100 mM NaCl, 0.1% Triton X-100, 300 mM Sucrose, 1 mM MgCl$_2$, 1 mM EGTA, 10 mM PIPES (pH 6.8; with NaOH) supplemented to contain 1x concentration of *Roche cOmplete Protease Inhibitor Cocktail EDTA-free)* and incubated for 10 min on 4˚C. To separate the soluble from the insoluble fraction, the suspension was centrifuged (4˚C, 2550 × *g* for 5 min). The supernatant was removed and the pellet was resuspended with CSK-buffer and centrifuged (4˚C, 2550 x g for 5 min). The pellet with the chromatin fraction was resuspended in Laemmli buffer supplemented with 2.5% β-mercaptoethanol and protease inhibitor cocktail. Proteins were denatured at 90˚C for 10 min. For detection of H2A.Z and H3 a polyclonal affinity-purified H2A.Z rabbit antibody (kindly provided by Nicolai Siegel [54]) and a polyclonal H3 rabbit antiserum were used [91], respectively.

## Co-immunoprecipitation

Prior to the immunoprecipitation (IP), 30 μl of Protein G Sepharose Fast Flow beads (GE Healthcare) were washed with 1 ml phosphate-buffered saline (PBS) (4˚C, 1000 x g for 1 min) and twice in PBS/1% bovine serum albumin (BSA). Unspecific binding sites were blocked by incubation with PBS/1% BSA for 1 h at 4˚C on an orbital mixer. The beads were pelleted by centrifugation (4˚C, 500 x *g* for 1 min), supernatant removed, and the antibody diluted in PBS added. Beads and antibody were incubated overnight at 4˚C. Unbound antibody was removed by washing 3x in 1 ml PBS/0.1% BSA. Before adding the lysate for immunoprecipitation, the beads were washed with 1 ml IP-Buffer buffer (150 mM NaCl, 0.5% IGEPAL CA-630, 20 mM Tris–HCl, pH 8.0, 10 mM MgCl$_2$, 1 mM dithiothreitol (DTT), protease inhibitor cocktail (*Roche cOmplete*)) and subsequent centrifugation (at 4˚C, 1000 x g for 1 min). Per IP, 1x10$^8$ cells were harvested and washed with ice-cold TDB (BSF), or with PBS (PCF), and incubated with 1 ml IP buffer for 20 min on ice. Cells were lysed by sonication (5 cycles, each 30 s on and 30 s off) using a Biorupter (Diagenode). A centrifugation step (at 4˚C; 20.000 x *g* for 15 min) followed to separate the soluble from the insoluble fraction. The soluble fraction was then added to the Protein G sepharose beads coupled to either HA 12CA5 mouse monoclonal IgG (Sigma Aldrich) or anti-Ty1 (BB2) mouse monoclonal IgG [91] and incubated at 4˚C for 3 h. Beads were washed two times with 1 ml IP buffer for 10 min at 4˚C. Proteins were eluted by incubating the beads in 50 μl sample buffer (1xNuPAGE LDS Sample buffer with 100 mM DTT) at 70˚C for 10 min. Eluates were then analysed by mass spectrometry (MS).

## Label-free quantitative mass spectrometry analysis

Samples were run on a Novex Bis-Tris 4–12% gradient gel (Thermo) with '3-(N-morpholino) propanesulfonic acid' (MOPS) buffer (Thermo) for 10 min at 180 V. The gel was stained with Coomassie blue G250 dye (Biozym) prior to cutting each gel lane into pieces, the gel lanes were minced and destained in 50% EtOH/water. The gel pieces were dehydrated with pure acetone, reduced with 10 mM DTT (Sigma Aldrich) and alkylated with 55 mM iodoacetamide (Sigma Aldrich) in the dark. The dried gel pieces were rehydrated with 1 μg trypsin for an in-gel digestion overnight at 37˚C. On the following day the digested peptides were desalted and stored on StageTips [104] for further analysis. Using a C18 reverse phase column that was previously packed in-house with Reprosil C18 (Dr. Maisch GmbH) the peptides were separated along a 105 min gradient using an EasyLC 1000 UHPLC system. The column was enclosed into a column oven (Sonation) and peptides were sprayed into a Q Exactive Plus mass spectrometer (Thermo), which was operating in a data-dependent top 10 acquisition mode. Spray voltage was set to approximately 2.4 kilovolt (kV). The acquired raw files were processed with MaxQuant (version 1.5.8.2) [105] using the *Trypanosoma brucei* protein database downloaded

from TriTrypDB and activated LFQ quantitation. Contaminants, reverse hits and protein groups that were only identified by site and protein groups with less than two peptides (one of them unique) were removed prior to bioinformatics analysis. For enrichment, the median of the log2 LFQ intensity values of the replicates was calculated and the p-value was determined by a Welch t-test between the IP and the control sample. The volcano plot was plotted using the R environment. The MS proteomics data have been deposited to the ProteomeXchange Consortium via the PRIDE partner repository with the data identifier PXD026796. The MS data are summarized in S2 Table.

## Western blot analysis and antibodies

Protein extracts of $3x10^6$ to $8x10^6$ cells were separated on 15% sodium dodecyl sulphate-polyacryl-amide gel electrophoresis (SDS-PAGE) gels and transferred onto polyvinylidene difluoride (PVDF) membranes. Membranes were blocked in 5% milk powder in PBS at 4˚C overnight. Primary antibodies were applied in PBS/1% milk/0.1% Tween-20 solution for 1 h at RT. After three washes with PBS/0.1% Tween 20, IRDye 800CW- and 680LT-coupled secondary antibodies (LI-COR Biosciences) were used to detect the corresponding primary antibodies. Secondary antibodies were incubated in PBS/1% milk/0.1% Tween20/0.02% SDS solution for 1 h at RT. After the incubation, blots were washed with PBS/0.1% Tween 20. Blots were analysed using a LI-COR Odyssey Imager (LI-COR Biosciences). Images were quantified with the Image Studio Lite Software. Background subtraction was performed by defining areas for normalization above and below the measured area. The polyclonal anti-TbH3 rabbit antibody was described in [106]. The monoclonal mouse BB2 anti-Ty1 antibody was derived from a hybridoma cell line and described in [91]. The monoclonal mouse anti-HA 12CA5 antibody was obtained from Sigma Aldrich.

## Cell cycle analysis

$5x10^6$ BSF cells were centrifuged (at 4˚C, 1.500 x g for 10 min) and washed once with 5 ml ice-cold TDB. After subsequent centrifugation (at 4˚C, 1.500 x g for 10 min) the cells were resuspended in 1 ml ice cold PBS/2 mM EDTA and fixed by dropwise adding 2.5 ml ice cold 100% ethanol or methanol. Cells were fixed at 4˚C for 1 h. After centrifugation (at RT, 1.500 x g for 10 min) cells were washed with 1 ml PBS/EDTA, centrifuged (at RT, 1.500 x g for 10 min) and resuspended in 1 ml PBS/EDTA. To the suspension 1 μl RNase (10 μg/μl) and 10 μl propidiumiodide (1 μg/μl) were added and incubated for 30 min at 37˚C. Samples were stored at 4˚C in the dark until analysis with a BD FACSCalibur using the FL-2 detector channel.

## Live/dead staining

$1x10^6$ BSF cells were centrifuged (at 4˚C, 1.500 x g for 10 min) and washed twice with 1 ml ice-cold TDB (stored on ice). After each washing step, the cells were centrifuged (at 4˚C, 1.500 x g for 10 min). The cells were resuspended in 400 μl TDB and incubated with 1 μl propidium iodide (1 mg/ml with a final concentration of 2.5 μg/ml) for 10 min on ice, in the dark. After the staining, the cells were analysed with a BD FACSCalibur.

## Luciferase assay

Luciferase assays were performed using the dual luciferase assay system (Promega). $1x10^6$ cells were centrifuged (1500 x g for 10 min at 4˚C) and washed with ice-cold PBS. After centrifugation (at 4˚C, 1.500 x g for 10 min) the supernatant was discarded and the cells were resuspended in 155 μl passive lysis buffer. The 155 μl were immediately transferred into a transparent flat bottom 96-well plate. 55 μl of stop&glow solution were added in the dark and

the lid was covered to prevent light incidence. The plate was incubated for 2 minutes at RT. The samples were analysed with an Infinite 200M plate reader (TECAN). Each well was measured for a duration of 1s.

### Northern blot analysis

Northern blots were done as previously described [107]. mRNA was prepared with the RNeasy kit (Qiagen). 18S rRNA and 5.8S rRNA were detected with antisense oligos coupled to IRDye 800, namely 5'-CCTTCGCTGTAGTTCGTCTTGGTGCGGTCTAAGAATTTC-3' and 5'-ACTTTGCTGCGTTCTTCAACGAAATAGGAAGCCAAGTC-3', respectively. Total mRNA and SL RNA were detected by an oligo antisense to the mini-exon sequence (5'-CAATATAG TACAGAAACTGTTCTAATAATAGCGTT-3'), coupled to IRDye 700. Blot images were obtained with the Odyssey Infrared Imaging System (LI-COR Biosciences) and quantified with the Image Studio Lite Software. Background subtraction was performed by defining areas for normalization above and below the measured area (for 5.8S rRNA) or manually by defining a square in between the lanes (total mRNA).

### EM sample preparation and imaging

The EM sample preparation protocol can be found in [108] and was adapted from [109–111]. $3x10^7$ BSF (bloodstream form) cells were centrifuged (750 g, 3 min, RT). All but 2 ml medium were removed and 2 ml heat-inactivated fetal calf serum was added as a cryoprotectant. Cells were centrifuged (750 g, 3 min, RT) and the pellet was transferred to a polymerase chain reaction (PCR) tube and further compacted (10 s, minifuge). A drop of the final pellet (around 1.5 μl) was transferred to the freezing container (specimen carriers type A, 100 μm, covered with specimen carriers type B, 0 μm, Leica Microsystems). High pressure freezing was done in an EM HPM100 (Leica Microsystems) at a freezing speed >20 000 $Ks^{-1}$ and a pressure >2100 bar. The samples were stored in liquid nitrogen until freeze substitution in an EM AFS2 freeze substitution system (Leica Microsystems).

For embedding in Epon, samples were incubated in 0.1% (w/v) tannic acid and 0.5% (v/v) glutaraldehyde in anhydrous acetone at −90˚C for 96 h (with one change in solution after 24 h), washed four times for 1 h with anhydrous acetone at −90˚C and fixed in 2% $OsO_4$ (w/v) in anhydrous acetone at −90˚C for 28 h. Then the temperature was gradually raised to −20˚C within 14 h, kept at −20˚C for 16 h and gradually raised to 4˚C within 4 h. Afterwards samples were immediately washed with anhydrous acetone at 4˚C four times at 0.5 h intervals, followed by gradually increasing the temperature to 20˚C within 1 h. Subsequently, samples were transferred for embedding into increasing concentrations of Epon (50% Epon in acetone for 3 h at room temperature, 90% Epon in acetone overnight at 4˚C, followed by two times 100% Epon at room temperature for 2 h, all solutions were freshly prepared). Epon infiltrated samples were polymerized for 72 h at 60˚C.

For staining and contrasting, Epon-embedded sections were incubated in 2% aqueous uranyl acetate for 10 min followed by incubation in Reynolds lead citrate for 5 min. LR White-embedded sections were incubated in 2% aqueous uranyl acetate for 5 min followed by incubation in Reynolds lead citrate for 1.5 min. A 200 kV JEM-2100 (JEOL) transmission electron microscope or a 30 kV JEOL JSM-7500F Scanning Electron Microscope equipped with a Tem-Cam F416 4k x 4k camera (Tietz Video and Imaging Processing Systems) was used for imaging.

### Supporting information

**S1 Table. A Table. Identification of a new complex associated with HAT2**. 10 proteins were identified via mass spectrometry in two Co-IP experiments. The initial Co-IP was performed

with Tb927.11.11530, the reciprocal Co-IP with the protein Tb927.11.10070 was performed to confirm the Tb927.4.2000 co-IP data. Only Tb927.3.4140 could be identified in the initial but not in the reciprocal Co-IP experiment. The "Annotation" column indicates the curated annotation that was found for the corresponding accession number in the TriTyp database. The" identified domains" column displays the domains that were found by BLAST search using the NCBI database. The Phyre2 modelling column indicates proteins that were identified by homology modelling. Coverage (Cov.) indicates the coverage in percent between query and template. The confidence (Conf.) represents the relative probability in percent (from 0 to 100) that the match between query and template is a true homology. The nuclear enrichment score (NES) indicates a nuclear localization if positive. The last column shows in which of the two Co-IPs the protein could be identified. B Table. Identification of a new complex associated with HAT1. 11 proteins were identified via mass spectrometry in two co-IP experiments. The initial Co-IP was performed with Tb927.9.2910, the reciprocal Co-IP with the protein Tb927.1.650 was performed to confirm the Tb927.9.2910 co-IP data. Tb927.6.1240, Tb927.10.8310 and Tb927.10.9930 could only be identified in the initial but not in the reciprocal co-IP experiments. The "Annotation" column indicates the curated annotation that was found for the corresponding accession number in the TriTyp database. Proteins labelled in green exhibit a homologue in the S. cerevisiae NuA4 complex. The nuclear enrichment score (NES) indicates a nuclear localization if positive. The "ident. in co-IP" column shows in which of the two Co-IPs the protein could be identified. The"identified domains" column displays the domains that were found by BLAST search using the NCBI / Interpro database. The Phyre2 modelling column indicates proteins that were identified by homology modelling. Coverage (Cov.) indicates the coverage in percent between query and template. The confidence (Conf.) represents the relative probability in percent (from 0 to 100) that the match between query and template is a true homology. The "Yeast NuA4 subunit"column states the NuA4 complex subunit with its corresponding domain ("domain(s)" column) to which the identified trypanosome protein is homologous to. C Table. Primer list. Primer sequences that were used for cloning of RNAi and knock out constructs and PCR amplification for in situ tagging.
(DOCX)

**S2 Table. Summary of all mass spectrometry data.** Excel file with original mass spectrometry data of TbRuvB IP (sheet 1), Tb4040 IP (sheet 2), TbYEATS IP (sheet 3), TbYL1 IP (sheet 4), TbEaf6 IP (sheet 5), TbEaf3 IP (sheet 6), TbHAT2 IP (sheet 7), and TbBdf3 IP (sheet 8).
(XLSX)

**S1 Fig. A Fig Trypanosoma brucei possesses a SNF2 protein with characteristics of the SWR1 subfamily.** A database search for SNF2 ATPases in T. brucei identified 15 proteins that putatively belong to the SNF2 superfamily. Only the protein Tb927.11.10730 has the characteristic structure of a protein of the SWR1 subfamily. For comparison, the structure of the SWR1 protein from S. cerevisiae is depicted (S.c. SWR1). In addition to the DEXQ DEAD-Box motif, a key feature of the SWR1 members of the SNF2 superfamily is an insertion (red box) between the DEAD-box and the helicase C domain. **B Fig Summary of volcano plots overview of 4 co-IPs.** Volcano blot of co-purified proteins after (A) WT control vs. HA-TbSWRC1 (Tb927.10.11690), (B) WT control vs. HA-TbSWRC2 (Tb927.11.5830), (C) WT control vs. TbSWRC4-HA (Tb927.7.4040) and (D) WT control vs. HA-RuvB2 (Tb927.4.2000), co-IPs obtained by MS analysis of four biological replicates. Green dots represent purified proteins with a p-value of $> 0.01$ or with a fold-enrichment of $= / > 1$. Blue dots represent purified proteins with a p-value $= / < 0.01$ or with a fold-enrichment of $> 1$. Orange dots represent the proteins with a p-value $= / < 0.01$ or with a fold-enrichment of $> 1$ that could be identified in

at least three of the four co-IPs. The annotations "measured" indicates that a sufficient number of unique peptides of the protein could be detected in the control samples to identify the corresponding protein. The annotation "some imputed" or "imputed" indicate that a theoretical value had to be imputed for some unique peptides that were used to identify the protein. **C Fig Depletion of TbSWR1 reduces the amount of chromatin associated H2A.Z**. (A) Growth of parasites was monitored for 96 hours after RNAi-mediated depletion of TbSWR1 (Tb927.11.10730) using tetraycline (tet). The parental 2T1 cell line was used as a control (n = 3). (B) Quantification of live/dead staining with propidium iodide of TbSWR1-depleted cells at the indicated timepoints post-induction. Analysis was done by flow cytometry (n = 3). (C) Western blot analysis of the insoluble nuclear fraction with antibodies specific for histone H3 and the histone variant H2A.Z. Lysates from an equal number of cells ($2x10^6$ per lane) were analysed for each timepoint. (D) Quantification of chromatin-associated H3 (dark blue), Ty1-H2A.Z (turquoise) and H2AZ (grey) (N = 3 for all depicted experiments; *** = p-value <0.001; ** = p-value 0.001–0.01; * = p-value 0.01–0.05). **D Fig Depletion of TbSWR1 leads to anucleate cell**. (A) Exemplary cell cycle profile of bloodstream form cells without (grey line) TbSWR1 (Tb927.11.10730) depletion and after 72 h of protein depletion (black line). The Gates show the different populations of sub G1-, G1-, S- and G2-Phase cells. (B) Data of three triplicates, sub G1 Phase cells (green), G1-Phase cells (green-blue), S-Phase cells (light blue) and G2-Phase cells (dark blue). The data show a decrease of cells in G1 and G2 Phase in addition to the increase of sub G1-Phase cells (n = 3 for all depicted experiments; *** = p-value <0.001; ** = p-value 0.001–0.01; * = p-value 0.01–0.05). (C) Light microscopy images (N = 1) of a BSF cell after 72h of TbSWR1 depletion. Scale bar 10μm. **E Fig Depletion of TbSWRC1 (Tb927.10.11690) reduces the amount of chromatin associated H2A.Z**. (A) Exemplary Western Blot analysis of the nuclear fraction with antibodies against histone H3 and the histone variant H2A.Z. An equal amount of cell equivalent was loaded for each timepoint. (B) The development of chromatin associated H3 (dark blue), Ty1-H2A.Z (green-blue) and H2A.Z (grey) in course of TbSWRC1 depletion is plotted (N = 3). (C) Growth of parasites was monitored for 96 hours after RNAi-mediated depletion of TbSWRC1 using tetracycline (tet). Growth of tet induced and non-induced parental 2T1 cells was measured for 96h and acts as a reference (N = 3 for all depicted experiments; *** = p-value <0.001; ** = p-value 0.001–0.01; * = p-value 0.01–0.05). **F Fig Depletion of TbSWRC2 (Tb927.11.5830) reduces the amount of chromatin associated H2A.Z** (A) Exemplary Western Blot analysis of the nuclear fraction with antibodies against histone H3 and the histone variant H2A.Z. An equal amount of cell equivalent was loaded for each timepoint. (B) The development of chromatin associated H3 (dark blue) and H2A.Z (turgoise) in course of TbSWRC2 depletion is plotted (N = 3). (C) Growth of parasites was monitored for 96 hours after RNAi-mediated depletion of TbSWRC2 using tetracycline (tet). Growth of tet induced and non-induced parental 2T1 cells was measured for 96h and acts as a reference (N = 3 for all depicted experiments; *** = p-value <0.001; ** = p-value 0.001–0.01; * = p-value 0.01–0.05). **G Fig H2A.Z acetylation pathway in S. cerevisiae and T. brucei**. Depiction of the H2A.Z acetylation pathway in S. cerevisiae (top panel) and T. brucei (bottom panel). In S. cerevisiae the NuA4 complex acetylates histone H4 to facilitate SWR1 recruitment to the nucleosome. SWR1 exchanges H2A with H2A.Z (10, 29, 30, 58). Subsequent to the exchange the NuA4 complex acetylates H2A.Z which enhances transcription (34, 41, 42). In T.brucei two distinct HAT-complexes are responsible for acetylation of histone H4 and histone H2A.Z. While H4 is the substrate for the HAT2 complex, H2A.Z is acetylated by the HAT1 complex. **H Fig Depletion of the histone acetyltransferase HAT1 caused a decrease of reporter luciferase activity within a PTU**. A single luciferase reporter construct was integrated into the tubulin array of a HAT1 (Tb927.7.4560) RNAi cell line. Samples for the luciferase assay were normalised to cell numbers. (A) Luciferase activity was

monitored for 48 h after induction of RNAi in two independent clones. Values of non-induced cells were set to 1. (B) Live/dead staining of each RNAi cell line was performed in triplicates at the same time points. (C) Growth of parasites was monitored for 96 hours after RNAi-mediated depletion of H2A.Z using tetracycline (tet) induction. Growth of the parental 2T1 cell line was measured for 96h as a control. (N = 3 for all depicted experiments; *** = p-value <0.001; ** = p-value 0.001–0.01; * = p-value 0.01–0.05). **1I Fig Depletion of the histone acetyltransferase HAT1 caused a decrease of reporter luciferase activity within a PTU**. A single luciferase reporter construct was integrated into the tubulin array of a HAT2 (Tb927.11.11530) RNAi cell line. Samples for the luciferase assay were normalised to cell numbers. (A) Luciferase activity was monitored for 48 h after induction of RNAi in two independent clones. Values of non-induced cells were set to 1. (B) Live/dead staining of each RNAi cell line was performed in triplicates at the same time points. (C) Growth of parasites was monitored for 96 hours after RNAi-mediated depletion of H2A.Z using tetracycline (tet) induction. Growth of the parental 2T1 cell line was measured for 96h as a control. (N = 3 for all depicted experiments; *** = p-value <0.001; ** = p-value 0.001–0.01; * = p-value 0.01–0.05). **J Fig Identification of a HAT2 complex**. Volcano blot of co-purified proteins after **(A)** Ty1-Bdf3 (Tb927.11.10070) vs. WT control **(B)** WT control vs. HA-HAT2 (Tb927.11.11530), co-IPs obtained by MS analysis of four biological replicates. Green dots represent purified proteins with a p-value of > 0.01 or with a fold-enrichment of = / > 1. Blue dots represent purified proteins with a p-value = / < 0.01 or with a fold-enrichment of > 1. The annotations "measured" indicates that a sufficient number of unique peptides of the protein could be detected in the control samples to identify the corresponding protein. The annotation "some imputed" or "imputed" indicate that a theoretical value had to be imputed for some unique peptides that were used to identify the protein. **K Fig Identification of a HAT1 complex**. Volcano blot of co-purified proteins after **(A)** WT control vs. Ty1-Bdf3 (Tb927.1.650) **(B)** WT control vs. HA-HAT2 (Tb927.9.2910), co-IPs obtained by MS analysis of four biological replicates. Green dots represent purified proteins with a p-value of > 0.01 or with a fold-enrichment of = / > 1. Blue dots represent purified proteins with a p-value = / < 0.01 or with a fold-enrichment of > 1. The annotations "measured" indicates that a sufficient number of unique peptides of the protein could be detected in the control samples to identify the corresponding protein. The annotation "some imputed" or "imputed" indicate that a theoretical value had to be imputed for some unique peptides that were used to identify the protein. **L Fig Chromatin condensation after *Tb*SWR1, H2A.Z and RNAP II (RPB1) depletion**. Representative electron microscopy images of the nucleus of *Tb*SWR1 **(A)**, H2A.Z **(B)** and RNAP II (RPB1; **C**) depleted cells. The upper panel depicts uninduced cell lines, the three panels below depicts TET induced cells (*Tb*SWR1 and H2A.Z: 24 h RNAi, RPB1: 16h RNAi). Depletion of the proteins resulted in large black patches of condensed chromatin. Scale bar, corresponds to 500 nm. Images obtained using a STEM (scanning transmission electron microscope). **M Fig Reduction in SWR1 mRNA molecules following RNAi depletion**. The number of SWR1 mRNA molecules per cell was measured by single molecule FISH. in the absence of TET and after 24 hours RNAi induction. Several unrelated mRNAs served as controls, as indicated. Affymetrix probe sets were designed antisense to the full ORF (*BDF3*), full ORF (*ZFP1*), 1100 most 5′nucleotides (Tb427.01.1730), full ORF (*CAF1*) and the repetitive sequence of *FUTSCH* (1). At least 46 cells were counted for each probe and timepoint. The data are presented as box-plots (waist is median; box is interquartile range (IQR); whiskers are 1.5 IQR). The differences in mRNA numbers in the absence and presence of SWR1 RNAi were evaluated by a students t-TEST (two samples, two-tailed; * for <0.05 = significant; *** for <0.005 = highly significant). There was a highly significant reduction in the number of SWR1 mRNA molecules upon RNAi induction. Note that two of the four control mRNAs also showed a significant reduction in numbers: this is expected from the

general reduction in transcription that is a consequence of SWR1 depletion.
(DOCX)

## Acknowledgments

We thank Brooke Morriswood for providing plasmids and helpful comments. Publication was supported by the Open Access Publication Fund of the University of Wuerzburg.

## Author Contributions

**Conceptualization:** Falk Butter, Christian J. Janzen.

**Data curation:** Albert Fradera Sola.

**Funding acquisition:** Falk Butter, Christian J. Janzen.

**Investigation:** Tim Vellmer, Laura Hartleb, Susanne Kramer, Elisabeth Meyer-Natus.

**Resources:** Falk Butter.

**Software:** Albert Fradera Sola.

**Supervision:** Falk Butter, Christian J. Janzen.

**Writing – original draft:** Tim Vellmer.

**Writing – review & editing:** Falk Butter, Christian J. Janzen.

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
