## [Decision Letter · Decision Letter 0]

9 Jun 2021

Dear Janzen,

Thank you very much for submitting your manuscript "A novel SNF2 ATPase complex in Trypanosoma brucei with a role in H2A.Z-mediated chromatin remodelling" for consideration at PLOS Pathogens. As with all papers reviewed by the journal, your manuscript was reviewed by members of the editorial board and by several independent reviewers. In light of the reviews (below this email), we would like to invite the resubmission of a substantially-revised version that takes into account the reviewers' comments.

Of particular significance are reviewer-2's comments questioning whether the data presented are sufficient to justify the conclusion that the SWR1 homologues identified form a genuine complex in *T. brucei*. They also ask why the *T. brucei*-specific proteins were not targeted for RNAi and the affect on H2AZ deposition assessed. Their comments have implications for the title of the manuscript and the novelty of the reported findings and should be fully addressed.

All three reviewers noted a range of other issues that should be addressed, including the need to deposit all mass spec and sequence data in appropriate public repositories, which should be linked to from the manuscript. Complete non-selective data summaries should also be included in the manuscript, as per comments by reviewers-1 and -2.

We cannot make any decision about publication until we have seen the revised manuscript and your response to the reviewers' comments. Your revised manuscript is also likely to be sent to reviewers for further evaluation.

Please submit your revised manuscript within 60 days if possible. If you anticipate a delay, please let us know the expected resubmission date by replying to this email. Please note that revised manuscripts received after the 60-day due date may require evaluation and peer review similar to newly submitted manuscripts.

Sincerely,

Sam Alsford

Guest Editor

PLOS Pathogens

David Horn

Section Editor

PLOS Pathogens

Kasturi Haldar

Editor-in-Chief

PLOS Pathogens

orcid.org/0000-0001-5065-158X

Michael Malim

Editor-in-Chief

PLOS Pathogens

orcid.org/0000-0002-7699-2064

Thank you for your submitted manuscript, which has been reviewed by three experts in the field. Their comments highlight a range of concerns.

Of particular significance are reviewer-2's comments questioning whether the data presented are sufficient to justify the conclusion that the SWR1 homologues identified form a genuine complex in T. brucei. They also ask why the T. brucei-specific proteins were not targeted for RNAi and the affect on H2AZ deposition assessed. Their comments have implications for the title of the manuscript and the novelty of the reported findings and should be fully addressed.

All three reviewers noted a range of other issues that should be addressed, including the need to deposit all mass spec and sequence data in appropriate public repositories, which should be linked to from the manuscript. Complete non-selective data summaries should also be included in the manuscript, as per comments by reviewers-1 and -2.

Reviewer's Responses to Questions

**Part I - Summary**

Reviewer #1: In this paper the authors have identified trypanosome chromatin remodelling complexes in the T. brucei by mass spectrometry, and shown that depletion of a component leads to a decrease in steady-state mRNA and deposition of histone H2AZ.

Reviewer #2: The chromatin remodeling complex SWR1 and its involvement in replacing canonical histone 2A in nucleosomes with the variant H2A.Z is conserved across the eukaryotic kingdom. Since H2A.Z is also found in Trypanosoma brucei, being specifically enriched in transcription initiation regions, Vellmer et al. set out to characterize trypanosome components of the SWR1 remodeling complex. Using co-IP experiments, they identified several conserved and putative SWR1 subunit homologs which might form a complex in trypanosomes (and the same is true for the HAT2 co-purified proteins). However, there is no additional evidence (second criterion) that these proteins indeed form the complex envisioned by the authors. The complex drawn in Fig. 1A is at best a model of a putative trypanosome complex and the work an initial characterization of SWR1 complex homologs, not more. The title therefore is an overstatement and even more so the statement (lines 195/6): “The 13 proteins that were identified in at least three of the four co-IP experiments assemble the trypanosome SWR1-like complex”. There is no assembly assay whatsoever.

While the authors present convincing data that trypanosome SWR1, as its eukaryotic counterparts, is essential for H2A.Z abundance and deposition, the study is mainly confirmatory in nature in this respect. Identifying SWR1 in the genome database should have been straightforward since it is by far the most conserved homolog to yeast SWR1 (E=e-99 versus e-64 to the next potential homolog). Moreover, the knockdowns of the study were restricted to clear homologs of yeast proteins and did not include putative trypanosome-specific genes, making their impact on H2A.Z predictable. Hence, this work provides only moderate progress in understanding trypanosomatid-specific SWR1 complex features. In addition, there are several concerns listed below.

Reviewer #3: This manuscript describes a series of co-immunoprecipitation experiments that identify 13 proteins forming an SWR1-like chromatin remodeling complex in Trypanosoma brucei and a serious of RNAi experiments that show it plays a critical role in H2A.Z incorporation at transcription start sites. With a few exception 9some of which are noted below), the manuscript is well-written, and the results will be of interest to most readers. However, a few (relatively minor) issues need to be addressed.

**Part II – Major Issues: Key Experiments Required for Acceptance**

Reviewer #1: 1. Deposit the proteomics and sequence data in a public database and give the accession numbers

2. Provide an Excel Table as a supplement for the quantitative MS.

3. All new gene assignments must be annotated using user comments in TritrypDB.

4. When there are only three measurements (which is everywhere) please replace bar graphs and standard deviations with plots of the individual values. For a only three measurements SDs are invalid, and almost never really reflect the distributions of the values.

5. Please provide a very approximate estimate of how much H2A.Z and pol II are left after RNAi - the existing Westerns are enough to do this.

6. Figure 8: The authors state that not all cells had chromosome condensation. Some quantitation is needed for this. How many showed it in + and - RNAi? Suggest sources of the variation.

7. Tables 1 and 2: provide data about localization from Tryp-Tag.

8. Line 118-119. Please delete the phrase “one of the earliest-branching eukaryotic lineages”. ALL eukaryotes belong to “early-branching” lineages. See 10.1101/cshperspect.a016147 or even 10.1073pnas.0807880106. As an alternative, authors could replace this phrase with an accurate statement.

9. There is no real need to defend use of luciferase as a reporter (lines 262-264) although and citing just one paper doesn’t demonstrate “extensive” use. It is really good for looking at initial gene expression. However it is really not appropriate to use it to measure decreases in mRNA since luciferase protein is, if wild-type, imported into the glycosomes and probably has quite a long half-life. Since this is - as the authors admit - a very indirect measurement, I think these measurements should be deleted. They do not really add to the paper since the mRNA measurements, which the authors also have done, are much better.

10. Line 398 - the lack of an effect on SLRNA is surprising and deserves a bit more comment. Could it be that the very particular pol II intiation complex that transcribes this - with its associated transcription factors - enables it to continue without SWR1? (Refer to relevant literature.) Also this is a steady-state level, and loss of trnascriuption might be compensated by the fact that there is less precursor available, so the SLRNA won’t be consumed as fast as usual.

Reviewer #2: Concerns

1) As stated in the title, the authors claim to have characterized ‘A novel SNF2 ATPase complex’. However, co-IP experiments alone are insufficient because [reciprocal] co-IP/-purification alone is not a stringent enough criterion for this claim (see also point 2). There is no additional evidence that these proteins do form a complex, meaning a stable entity, in trypanosomes. For example, the authors present a putative complex in Figure 1A (modeled according to the yeast counterpart) which probably has a combined molecular mass of several hundred kDa or more. If this complex does indeed exist, the authors need to present evidence (sizing column, native gel, sedimentation, etc) that some of the co-IPed proteins migrate together at the expected complex size. The same holds true for the HAT2 complex.

2) The presented mass spec data of the co-IPs are problematic. It is highly unusual to detect enrichment of only ~15 proteins in a co-IP with moderate E values. There are numerous reports in the field in which proteins were isolated in multiple steps in much higher purity than can be achieved by a single IP. Nevertheless, in these reports often more than a hundred proteins co-purified with enrichment and e-values that meet the criterions used here, and most of them did not define a complex. Hence, the data set presented appears to be highly selective. While I am not sure about PLoS Pathogens policy, mass spec data including “raw files” of all identified proteins, i.e. prior to the authors’ selection, should be deposited in public repositories such as PRIDE. Apparently, this has not been done and, therefore, it is impossible to assess whether the presented proteins in the tables were selected in an unbiased manner. In addition, the authors claim to have carried out “quantitative proteomics” (line 130). I assume they used label-free quantification; if so, this should be better specified in the methods.

3) The authors’ claim of a novel SWR1 (and HAT2 complex) is based on co-IP of proteins that are either conserved only in trypanosomes or not part of the corresponding complex in yeast. However, in addition to not using a second criterion for complex formation, the authors fail to show that these genes/proteins have H2A.Z-specific functions. How do they justify ‘novel’?

4) Figures 5-7: luciferase (protein-based) reporter assays and steady-state mRNA quantification are indirect methods to assay for a transcription defect. Transcriptional activity is best measured by labeling and quantifying newly synthesized pre-m/rRNA.

5) It should be mentioned in the results section how the proteins were tagged and which antibody and protein solution was used for the IP since this is critical information to evaluate co-IP & mass spec results. In addition, it would be reassuring if the authors showed immunoblots, demonstrating IP efficiency. (Probably, protein staining would not reveal enrichment).

6) Figure 8: is there a way to quantify condensed chromatin in un- versus treated cells to make this more convincing?

Reviewer #3: None noted

**Part III – Minor Issues: Editorial and Data Presentation Modifications**

Reviewer #1: Line 430 - it would be interesting to look at other eukaryotic supergroups too. What is known about plants?

line 46: elementary knowledge, delete. rest of paragraph can be shortened.

line 64: Should be “proteins of all subfamilies”.

Line 103: why “eponymous”? Also, need a comma after ATPase and another “which” instead of “that”. (“Which” describes, “that” defines; they can’t be interchanged.)

Lines 105- onwards - refer to Fig 1. I found Fig1 difficult to read - please make all the text at least the same size as the largest in the external labels, it appears to me that the labels “within” the proteins are currently smaller.

Last paragrpah of the introduction is a repetition of the abstract, and could be shortened just to state the aims.

Table 1 - what does the asterisk mean? Please explain briefly in the Legend how the NES scores can be interpreted (are they explained more in the MEthods?) Also mention how were the p-values obtained, and what the nerichemtn was relative to (i.e. what the control was). Decide whether or not to capitalise the first letter of the Annotation.

Fig 5 - unless you measured the copy-number, delete “single” from the legend since we can never be sure there was only one integration. (Same for FIg 6.) Labels on the graphs in Fig 5 are microscopic - and the superscripts can be replaced if you use “log cell number” as the label for the y-axis.

Fig 7: replace hairlines in “B” with slightly thicker lines.

Line 539: missing paragraph mark.

Lines 364-372 - I think these belong in the REsults section.

Fig S7 - should be “anucleate cells”. or better just “cells without nuclei”.

Line 387 - I believe that PLoS doesn’t allow the phrase “data not shown”. You can get around this by instead writing “preliminary results suggested that….”.

Line 399: “spliced leader mRNA that appears to be mainly unaffected by TbSWR1 depletion” implies that there is other spliced leader RNA that IS affected. REplace “that” with “which”.

Reviewer #2: point 5 above is a minor issue.

Reviewer #3: Line 188-189: Do the five proteins identified in the RuvB2 co-IP that are not part of the TbSWR1 complex have any similarity to the components of the yeast INO80 (or any other chromatin modification) complex?

Line 246: Please provide a reference for the Co-IP experiments with HAT2.

Line 224-335: What methods were used to exclude the presence of a DAMP-1 homologue in T. brucei?

Line 427: Is it possible that H2A.Z does not need to be removed from chromatin, since transcription initiation is essentially constitutive in trypanosomatids?

Line 428-429: Jensen et al (mSphere, 2021) recently described a putative chromatin remodeling complex associated with JBP3 in Leishmania, orthologues of which are present in T. brucei.

Would it be possible to combine Tables 1 and 2?

Lines 1026 and 1031: “bolt” should be replaced with “bold” in the Figure 1legend.

Figure 3: As currently shown, the depletion of H2A.Z at TSSs is not very convincing. I suggest that using more contrasting colors and/or normalizing to WT might be helpful, as would adding cartoons representing the different PTUs in panel A.

PLOS authors have the option to publish the peer review history of their article (what does this mean?). If published, this will include your full peer review and any attached files.

Reviewer #1: No

Reviewer #2: No

Reviewer #3: **Yes: **Peter J Myler
---

## [Decision Letter · Decision Letter 1]

1 Dec 2021

Dear Prof Janzen and Prof Butter,

Thank you very much for submitting your manuscript "A novel SNF2 ATPase complex in Trypanosoma brucei with a role in H2A.Z-mediated chromatin remodelling" for consideration at PLOS Pathogens. As with all papers reviewed by the journal, your manuscript was reviewed by members of the editorial board and by several independent reviewers. The reviewers appreciated the attention to an important topic. Based on the reviews, we are likely to accept this manuscript for publication, providing that you modify the manuscript according to the review recommendations.

Before we can make a final decision, please can you address the following outstanding issues:

1. Adjust the data presentation as per reviewer-1's comment (part-III)

2. Clarify your response (2.1) to reviewer-2's comments on confirmatory data for the putative novel complex. You mentioned that you've run native gels following coIP, but not been able to detect protein by silver staining. Have you been able to detect the complex on these native gels using anti-tag antibodies? Are you able to localise (or co-localise) the tagged proteins, rather than citing TrypTag procyclic data?

In addition to the above and the reviewer comments below, please can you address the following:

3. Provide evidence (northern or qRT-PCR) for SWR1 and Bdf3 knockdown

4. Update the Staneva ref to reflect its recent publication in Genome Research

5. Remove SK from the acknowledgements (already an author)

6. Two versions of the figures were included in your revised submission - please ensure that only the definitive versions are included in future; one version of figures 5 and 7 includes the correct sub-panel labeling, the other doesn't

7. In the chart legends, please be consistent in the use of 'light blue' or 'turquoise'; if using the latter, please check the spelling (S9)

Sincerely,

Sam Alsford, Ph.D.

Guest Editor

PLOS Pathogens

David Horn

Section Editor

PLOS Pathogens

Kasturi Haldar

Editor-in-Chief

PLOS Pathogens

orcid.org/0000-0001-5065-158X

Michael Malim

Editor-in-Chief

PLOS Pathogens

orcid.org/0000-0002-7699-2064

Dear Prof Butter and Prof Janzen,

Thanks for your revised manuscript. Before we can make a final decision, please can you address the following outstanding issues:

1. Adjust the data presentation as per reviewer-1's comment (part-III)

2. Clarify your response (2.1) to reviewer-2's comments on confirmatory data for the putative novel complex. You mentioned that you've run native gels following coIP, but not been able to detect protein by silver staining. Have you been able to detect the complex on these native gels using anti-tag antibodies? Are you able to localise (or co-localise) the tagged proteins, rather than citing TrypTag procyclic data?

In addition to the above (and the specific reviewer comments below), please can you address the following:

3. Provide evidence (northern or qRT-PCR) for SWR1 and Bdf3 knockdown

4. Update the Staneva ref to reflect its recent publication in Genome Research

5. Remove SK from the acknowledgements (already an author)

6. Two versions of the figures were included in your revised submission - please ensure that only the definitive versions are included in future; one version of figures 5 and 7 includes the correct sub-panel labeling, the other doesn't

7. In the chart legends, please be consistent in the use of 'light blue' or 'turquoise'; if using the latter, please check the spelling (S9)

Reviewer Comments (if any, and for reference):

Reviewer's Responses to Questions

**Part I - Summary**

Reviewer #1: In this paper the authors have identified trypanosome chromatin remodelling complexes

in the T. brucei by mass spectrometry, and shown that depletion of a component leads to a decrease

in steady-state mRNA and deposition of histone H2AZ.

Reviewer #2: The revised manuscript by Vellmer et al. has met several formal requirements of reviewers such as data submission to public data banks and changes in text. However, key experimental critiques could not be addressed. In general, the authors were unable to detect subunits of their complexes by tagging and to ablate the genes of putative trypanosome-specific subunits, rendering their [functional] participation in SNF2 and HAT2 complexes weakly supported. As detailed in the previous review, this study is mainly confirmatory in nature.

In sum, this study clearly shows that, as in other organisms, SWR1 is essential for H2A.Z deposition and it suggests that SWR1 ablation leads to a transcriptional defect. Although the latter was shown in a rather indirect manner (reporter protein), it is the most important aspect of the study.

However, the conceptual/experimental weaknesses of the study predominate and are reiterated here in short:

(i) Annotation of the SWR1 homolog in trypanosomes would have been straightforward by a simple BLAST search; it did not need the elaborate bioinformatics analysis laid out here.

(ii) The putative trypanosome SNF2 complex depicted in Fig 1 remains extremely speculative without a second criterion and any functional data supporting it. Given that dozens of gene expression factors have been readily detectable in various laboratories, it is difficult to understand why the authors were not able to detect them.

(iii) “Novel” features of the SNF2 complex (in the title!) are solely based on bioinformatics of co-precipitated proteins. There is no functional data supporting any of the authors’ claims. Again, the assertion of the authors that they tried in vain to ablate genes of trypanosome-specific subunits is not encouraging and speaks against the relevance of the encoded proteins in the two complexes.

(iv) Knockdown data are provided only for SWR1 which is a highly conserved protein with known function as well as BDF3 who has already been described and linked to H2.AZ. These functional data are mainly confirmatory in nature (Why could the authors knock down these factors successfully? Is it possible that the other knockdowns did not reveal anticipated results?).

Finally, the reference to another bioRxiv paper lacking functional data is not helpful.

Reviewer #3: The authors have responded to most of the comments from the previous review and the manuscript is now ready for publication.

**Part II – Major Issues: Key Experiments Required for Acceptance**

Reviewer #1: None

Reviewer #2: This was detailed in the original critique. The authors claim that they have tried to do the key experiments without success.

Reviewer #3: None

**Part III – Minor Issues: Editorial and Data Presentation Modifications**

Reviewer #1: In their responses the authors expend a lot of effort in saying why they don't want to change most aspects of the manuscript. I have just one request:

"We are also critical about this point, but other reviewers requested exactly this kind of statistics in the past. We would prefer to keep the figures in this format." If you are critical then change it! Just because reviewers make mistakes there is no need to follow their advice. I have had no trouble whatsoever publishing graphs with individual values, there is an excellent argument for doing so and it is becoming increasingly common. If you insist on keeping bar graphs with invalid standard deviations, at least superimpose the individual values as clearly visible spots.

Reviewer #2: at this point, not relevant.

Reviewer #3: None

PLOS authors have the option to publish the peer review history of their article (what does this mean?). If published, this will include your full peer review and any attached files.

Reviewer #1: No

Reviewer #2: No

Reviewer #3: **Yes: **Peter J Myler

Figure Files:

Data Requirements:

Reproducibility:

References:

---

## [Editor Report · Decision Letter 2]

8 Apr 2022

Dear Prof Janzen,

We are pleased to inform you that your manuscript 'A novel SNF2 ATPase complex in Trypanosoma brucei with a role in H2A.Z-mediated chromatin remodelling' has been provisionally accepted for publication in PLOS Pathogens.

Best regards,

Sam Alsford, Ph.D.

Guest Editor

PLOS Pathogens

David Horn

Section Editor

PLOS Pathogens

Kasturi Haldar

Editor-in-Chief

PLOS Pathogens

orcid.org/0000-0001-5065-158X

Michael Malim

Editor-in-Chief

PLOS Pathogens

orcid.org/0000-0002-7699-2064

Dear Prof Janzen and Prof Butter,

Thanks for your revised manuscript and your considered responses to the reviewers' outstanding comments.
---

## [Editor Report · Acceptance letter]

3 Jun 2022

Dear Janzen,

We are delighted to inform you that your manuscript, "A novel SNF2 ATPase complex in Trypanosoma brucei with a role in H2A.Z-mediated chromatin remodelling," has been formally accepted for publication in PLOS Pathogens.

Best regards,

Kasturi Haldar

Editor-in-Chief

PLOS Pathogens

orcid.org/0000-0001-5065-158X

Michael Malim

Editor-in-Chief

PLOS Pathogens

orcid.org/0000-0002-7699-2064